# A dynamic and adaptive network of cytosolic interactions governs protein export by the T3SS injectisome

Andreas Diepold[1,†], Erdinc Sezgin[2], Miles Huseyin[1], Thomas Mortimer[1,†], Christian Eggeling[2] & Judith P. Armitage[1]

Many bacteria use a type III secretion system (T3SS) to inject effector proteins into host cells. Selection and export of the effectors is controlled by a set of soluble proteins at the cytosolic interface of the membrane spanning type III secretion 'injectisome'. Combining fluorescence microscopy, biochemical interaction studies and fluorescence correlation spectroscopy, we show that in live *Yersinia enterocolitica* bacteria these soluble proteins form complexes both at the injectisome and in the cytosol. Binding to the injectisome stabilizes these cytosolic complexes, whereas the free cytosolic complexes, which include the type III secretion ATPase, constitute a highly dynamic and adaptive network. The extracellular calcium concentration, which triggers activation of the T3SS, directly influences the cytosolic complexes, possibly through the essential component SctK/YscK, revealing a potential mechanism involved in the regulation of type III secretion.

[1] Department of Biochemistry, University of Oxford, South Parks Road, Oxford OX1 3QU, UK. [2] MRC Human Immunology Unit, Weatherall Institute of Molecular Medicine, University of Oxford, Headley Way, Oxford OX3 9DS, UK. † Present addresses: Department of Ecophysiology, Max Planck Institute for Terrestrial Microbiology, Karl-von-Frisch-Straße 10, 35043 Marburg, Germany (A.D.); The Francis Crick Institute, 1 Midland Road, Kings Cross, London NW1 1AT, UK (T.M.). Correspondence and requests for materials should be addressed to J.P.A. (email: judith.armitage@bioch.ox.ac.uk).

To survive and multiply within host organisms, bacteria have evolved a multitude of ways to interact with the host, including the direct injection of effector proteins into host cells. This is accomplished by the type III secretion system (T3SS), which is employed by numerous Gram-negative bacteria[1–3]. The T3SS, also called injectisome, is essential for virulence in many important human pathogens, including *Salmonella*, *Shigella* and pathogenic *E. coli*, that cause millions of deaths per year[4]. It also plays an important role in nosocomial infections, most prominently in *Pseudomonas aeruginosa*[5], where the presence of the T3SS effector ExoU is associated with high mortality in animal models, increased antibiotic resistance and a more severe disease in infected humans[6]. However, there are currently no broadly applicable and efficient T3SS inhibitors[7,8] and attempts to rationally design inhibitors are limited by our knowledge of the molecular function and regulation of the T3SS.

The injectisome is evolutionarily closely related to the bacterial flagellum[9–11], and is characterized by a hollow needle, which is anchored in the bacterial membranes by the 'basal body', a series of rings that span both membranes. Five integral inner membrane (IM) proteins, termed the 'export apparatus', are thought to form the export gate, enclosed by an IM ring. At the cytosolic interface of the basal body, four interacting proteins are essential for the function of the T3SS (Fig. 1a). An oligomerization-activated ATPase, SctN (YscN in *Yersinia*, see Supplementary Table 1), is thought to detach T3SS effector chaperones, and then unfold the effectors to facilitate export[12,13], although it is dispensable for export under specific conditions[14]. The ATPase interacts with a negative regulator, SctL/YscL[15]. In the flagellum, the equivalent negative regulator forms a soluble 2:1 complex with the ATPase[16]. SctQ/YscQ is the homologue of the flagellar C-ring proteins. Using an internal translation start site, bacteria express both the full-length protein, which shares homology with the large flagellar C-ring protein FliM, and the C-terminal fragment SctQ$_C$/YscQ$_C$[17,18], which is highly homologous to the small flagellar C-ring protein FliN. FliM and FliN are part of the 'switch complex', which reverses the rotation of the *E. coli* flagellar motor in chemotaxis. The injectisome is generally not believed to rotate, and the injectisome C-ring has been shown to bind export cargo, specifically the T3SS chaperones, both in *Chlamydia*[19] and *Salmonella*, where native gel experiments showed that a complex of Sct/YscKQL, but not SctN/YscN, binds chaperones with different affinities, suggesting that it determines the order of effector export by acting as a sorting platform[20]. SctK/YscK is the least studied of the four soluble T3SS components, and is the only protein with no clear flagellar homologue. SctK/YscK localizes to the membrane in *E. coli*, independent of the presence of other T3SS components, and is required for the efficient export of effectors[21], but its exact role in secretion remains unclear. *In vitro* experiments have established a chain of interactions Sct/YscK-Q-L-N[19,22,23], and we will refer to these four core proteins as 'soluble T3SS components'. Additional soluble proteins involved in type III secretion are exported by the system itself, such as SctI/YscI, the 'inner rod' protein interacting with the needle[24]; SctO/YscO, whose flagellar homologue interacts with the ATPase and the export apparatus[25,26]; and species-specific proteins like YscX in *Yersinia*, which also interacts with the export apparatus[27,28]. The absence of these proteins does not influence the assembly of the core soluble T3SS components[28,29], and they were therefore not included in this study.

The needle and basal body are structurally very well characterized[30,31]. Although the physical contact to host cells, which activates the system, appears to be sensed at the needle tip and signalled through needle and basal body to the cytosolic interface of the injectisome[32–34], this scaffold shows little structural change between the resting and secreting state of the T3SS (ref. 35). The molecular events that direct the selection and secretion of effectors therefore probably occur at the cytosolic interface of the T3SS. We are only just beginning to get an insight into the structural organization of these proteins—with the first results showing striking differences between the injectisome and the flagellum[36–38]. However, it has already become clear that at least one of the soluble components, the C-ring protein SctQ/YscQ, is not a stable part of the injectisome, but is in dynamic exchange between the proximal interface and a cytosolic pool[39].

The current knowledge of the structure and interactions of the cytosolic components is mainly based on *in vitro* studies, and provides a limited understanding of the selection and export of substrates, or regulation of the T3SS. We therefore studied the interactions and dynamics of the soluble components of the injectisome in live bacteria and monitored the changes upon activation of secretion using a range of tailored and complementary biochemical and microscopy methods together with single-molecule spectroscopy techniques. We show that a percentage of the soluble components form a large complex at the proximal interface of the injectisome, whose formation requires all of its members. However, a substantial proportion of these T3SS components is diffuse throughout the cytosol. These diffusing proteins interact with each other in live bacteria, even in the absence of the major IM ring component SctD/YscD. These stable cytosolic complexes do however depend on the T3SS ATPase SctN/YscN. Fluorescence correlation spectroscopy (FCS) revealed that *in vivo*, the cytosolic T3SS components are forming a dynamic and highly adaptive cytosolic interaction network. The diffusion rate of these soluble complexes was influenced by the extracellular $Ca^{2+}$ concentration, which governs the secretion state of the injectisome in Yersinia. As this occurred even in the absence of T3SS needles, our results indicate that $Ca^{2+}$ may directly influence the formation and structure of cytosolic T3SS complexes, revealing a direct pathway, independent of T3SS needles, for bacteria to react to the external conditions and activate the T3SS.

## Results

**Characterisation of the cytosolic complex of the T3SS.** To analyse the localization of the soluble components of the T3SS (Fig. 1a), we constructed genetically encoded fluorescent fusions to the cytosolic T3SS components in *Yersinia enterocolitica*. The wild-type (WT) genes on the virulence plasmid were modified by allelic exchange[40] to encode for N-terminally enhanced green fluorescent protein (EGFP)-tagged versions of the respective protein, and the functionality of the T3SS in the resulting strains was tested by analysing the proteins secreted into the culture supernatant under secreting conditions. All strains were functional for secretion (Supplementary Fig. 1a) and, with the exception of EGFP-YscN, which displayed a double band possibly caused by partial degradation, the fusion proteins were stable (Supplementary Fig. 1b). All fusion proteins were expressed throughout the bacterial population, with a low (<5%) proportion of non-fluorescent bacteria. We then analysed the localization of the cytosolic T3SS components under different conditions by fluorescence microscopy. All labelled components localized in foci at the bacterial membrane (Fig. 1b), as shown previously for the basal body components, as well as YscQ and YscN[29]. These 'injectisome'-type foci were present under secreting as well as non-secreting conditions (Fig. 1b and Supplementary Fig. 2). The foci for the different components colocalized with each other and the IM ring protein YscD, which forms part of the basal body (Fig. 1c and Supplementary Fig. 3), showing that the large majority of injectisomes possess complete cytosolic interfaces.

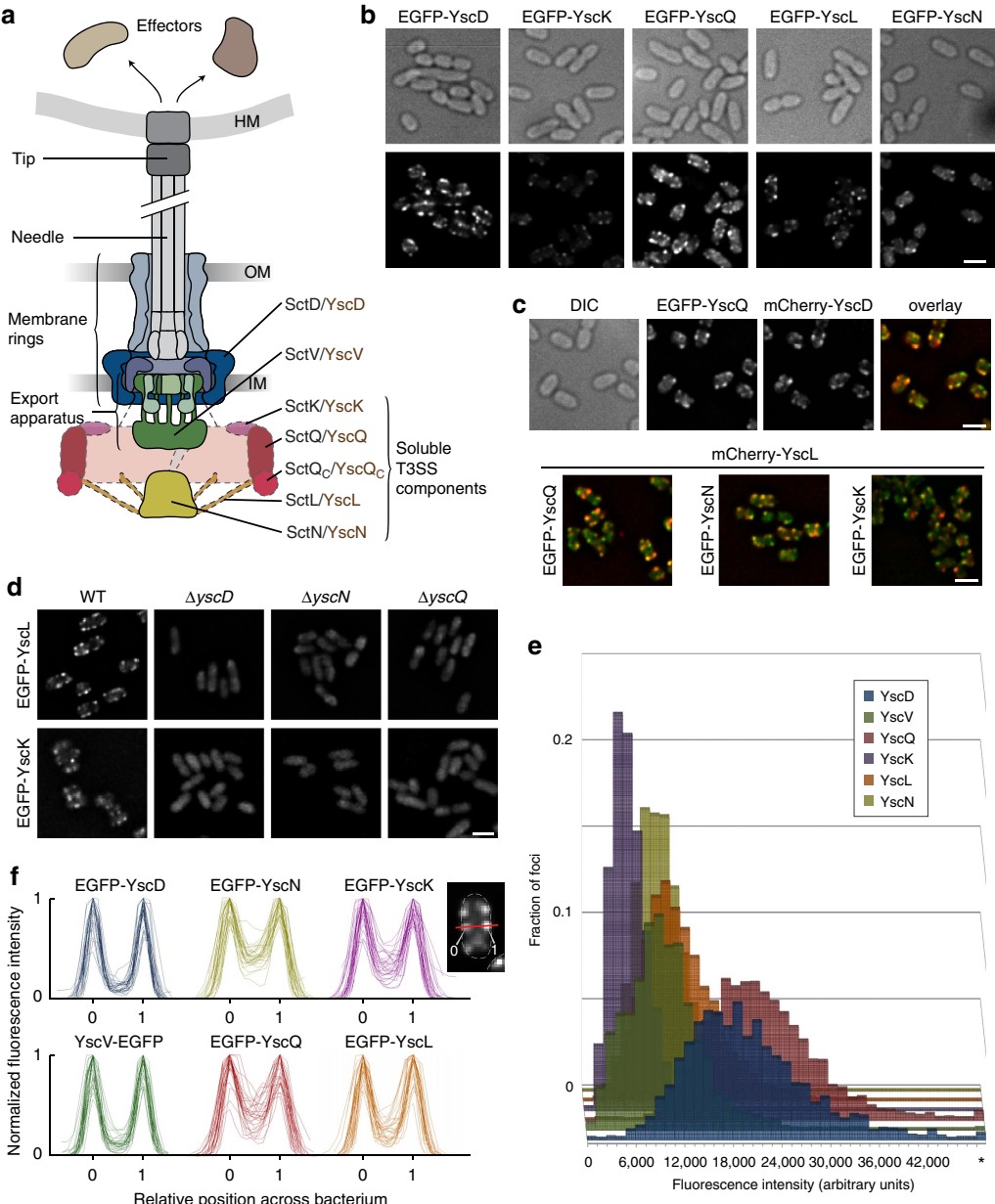

**Figure 1 | The soluble T3SS components form a complex at the cytosolic interface of the injectisome. (a)** Schematic overview of the T3SS injectisome (modified from ref. 71, based on cryoelectron microscopy data of purified needle complexes[30]). General (black) and *Yersinia*-specific (brown) names of the soluble T3SS components forming the cytosolic complex at the injectisome are indicated. (**b**). All soluble T3SS components localize in a similar manner in membrane foci, similar to the pattern of the membrane-ring component YscD. Differential interference contrast (DIC) images and respective deconvoluted GFP fluorescence micrographs of bacteria expressing N-terminal EGFP fusions of the respective proteins under non-secreting conditions (see Supplementary Fig. 2 for secreting conditions). The representative micrographs shown allow the comparison of fluorescence intensity. Each strain shown in **b**–**d** was imaged in three to six independent experiments yielding reproducible results. (**c**) The soluble components colocalize with each other and the basal body. DIC image, deconvoluted GFP and mCherry micrographs, and GFP/mCherry overlay of bacteria expressing the denoted labelled T3SS components under non-secreting conditions. Representative DIC and single colour GFP/mCherry fluorescence micrographs for the overlays displayed in the bottom row are shown in Supplementary Fig. 3. (**d**) The cytosolic complex requires each of its components to assemble at the T3SS. Micrographs of strains expressing EGFP-YscL or EGFP-YscK in a wild-type strain background or in the absence of YscD, YscN or YscQ. (**e**) Distribution of fluorescence intensities of foci in strains expressing the denoted T3SS components (C-terminal EGFP fusion for YscV, N-terminal EGFP fusions for all other proteins); 1,900 to 3,640 spots from 262 to 537 bacteria from six fields of view in two independent experiments per strain. (**f**) The soluble T3SS components YscK, YscQ, YscL and YscN are present both at the injectisome and in the cytosol of bacteria, whereas little cytosolic fluorescence is observed for the membrane components YscD and YscV. Line profiles of fluorescence crossing two fluorescent foci on opposite sides of the bacterium (see scheme) were generated for 32 bacteria from six fields of view in two independent experiments in deconvoluted micrographs of bacteria expressing N-terminal EGFP-fusions of the denoted proteins (C-terminal fusion in the case of YscV) were normalized for maximal fluorescence intensity (*y*-axis) and position along the bacterial diameter as indicated by the fluorescence maxima (*x*-axis, marks show position of the maxima) and overlaid. The scheme in the top right corner shows a representative line profile for a bacterium expressing EGFP-YscK. The red line marks the path of the intensity profile, the triangles and numbers note the intensity maxima used to normalize the diameter of the bacterium from relative position 0 to 1 and the shape of the bacterium is indicated by the dotted lines. Scale bars, 2 μm.

**Table 1 | Estimation of the relative stoichiometry of the soluble T3SS components.**

|  | Relative stoichiometry | Standard deviation | Literature value | References | Spots/bacterium |
|---|---|---|---|---|---|
| EGFP-YscD | **24.0**[*] | 7.8 | 24 | 42–45 | **8.6** |
| YscV-EGFP | **11.2** | 4.7 | 9 | 60 | **7.2** |
| EGFP-YscK | **6.7** | 2.2 |  |  | **6.8** |
| EGFP-YscQ | **27.6** | 7.2 | 22 ± 8 | 39 | **8.0** |
| EGFP-YscL | **14.2** | 4.7 | 12 | 16 | **7.1** |
| EGFP-YscN | **12.2** | 3.3 | 6/12 | 12,38,61 | **7.2** |

T3SS, type III secretion system.
Bacteria expressing the denoted fluorescently labelled proteins were imaged and the total fluorescence intensity of each spot of 1,900–3,640 detected spots from 262 to 537 bacteria per strain was used to estimate the relative stoichiometry of the labelled components at the injectisome, based on the reference of YscD, present in 24 subunits per injectisome. The resulting relative stoichiometry and average number of detected spots per bacterium are displayed in bold font.
*YscD, present in 24 copies, was used as reference for the stoichiometry of the other T3SS components.

To investigate the requirements for the recruitment of the different soluble components to the injectisome, we tested their localization in strains lacking other T3SS components. We found that all soluble components were required for the localization of YscK and YscL to the injectisome, as EGFP-YscK and EGFP-YscL were diffuse in the cytosol in the absence of any other member of the complex (Fig. 1d). Similarly, both proteins were diffuse in the negative control strains lacking YscD, an essential early factor in the assembly of the injectisome[29].

**Relative stoichiometry and distribution of T3SS components.**
To determine the occupancy and relative stoichiometry of the labelled components at the injectisome, we compared the intensity of the fluorescent foci in strains expressing different EGFP-labelled T3SS components[29,41] (Supplementary Figs 4 and 5). Foci were detected in three dimensions in unprocessed z-stacks of non-secreting bacteria using the Imaris software (Supplementary Fig. 5), and the mean fluorescence intensity for 1,900 to 3,640 detected spots per strain (Fig. 1e) was used to calculate the resulting relative stoichiometry (Table 1) on the basis of YscD, which is present in 24 copies[30,42–45]. The number of detected spots ranged from 6.8 to 8.6 per bacterium (Table 1). The slightly lower number of spots per cell for the low-copy number proteins YscK, L, N and V might be due to a low fraction of undetected weak foci, which would result in a slight overestimation of the stoichiometry of these components. Overall however, the similar number of spots per bacterium indicates that the majority of all injectisomes harbour cytosolic complexes containing all soluble T3SS components.

In contrast to the membrane-bound T3SS proteins, the soluble components shared a significant cytosolic background (Fig. 1f). We have shown previously that YscQ exchanges between its docked state at the injectisome and a free cytosolic pool[39]; therefore, this cytosolic pool might be essential for the function of the T3SS. We thus wanted to identify how the soluble T3SS components interact with each other in the cytosol, and whether this relates to function.

**Stable cytosolic complexes that are not bound to the injectisome.**
We hypothesized that the C-ring proteins, and perhaps other components of the T3SS, interact with each other in the cytosol when free of the injectisome. To test this hypothesis, we performed co-immunoprecipitations with functional Halo fusions to the central components of the cytosolic complex, YscQ and YscL (Supplementary Fig. 6). The quick and gentle purification procedure permits the detection of stable interactions occurring within the bacterium. As the Halo-tagged bait protein covalently binds to the resin and is not eluted, homo-oligomerization can be detected, through fusion proteins that interact but do not bind to the resin, as well as the formation of hetero-oligomers. We found that YscK, Q, L and N interact with each other, in agreement with

earlier studies[15,19,22,23,46,47]. These interactions were observed under secreting as well as non-secreting conditions (Table 2).

To analyse both the interaction network at the injectisome and in the non-injectisome-bound fraction, we performed the experiment in WT bacteria and in strains lacking YscD (Fig. 2a). Strikingly, stable interactions between the soluble components could be detected for all tested pairs in the Δ*yscD* strains (Fig. 2b), where all the studied components are diffuse throughout the cytosol (see Fig. 1d). In contrast to the cytosolic 'sorting platform' complexes observed by Lara-Tejero *et al.*[20], these complexes also included the ATPase YscN (Table 2 and Fig. 2b). In all cases, the degree of interaction was lower in the Δ*yscD* than in the WT strains, ranging from ∼20% for the YscQ–YscN and YscQ–YscK interactions to ∼50% for YscQ–YscL and the YscQ homo-oligomerization (Fig. 2b). Taken together, these data provide strong evidence for the formation of stable cytosolic complexes including all four soluble T3SS components YscK, YscQ, YscL and YscN. They also indicate that these complexes are less stable when not bound to the basal body of the injectisome, and probably have a more dynamic exchange under these conditions, especially YscK and YscN, the peripheral members of the complex.

The deletion of the ATPase YscN had a much more severe impact on the formation of the cytosolic complexes. The stable interactions between different soluble T3SS components were strongly decreased (YscQ–YscQ and YscQ–YscL) or not detectable (YscQ–YscK) (Fig. 2b). We tested the interaction between YscQ and its C-terminal fragment YscQ$_C$ by immuno-blot. While there is some reduction in the amount of interacting YscQ$_C$, this interaction appears to be the least affected by the absence of YscN (Fig. 2c and Supplementary Fig. 7), in agreement with earlier results[17,18,48]. Figure 2e summarizes the impact of YscD and YscN on the identified interactions.

**A dynamic and adaptive network of cytosolic interactions.** The results above showed that the soluble components form a network of interactions, both at the injectisome and within the cytosol.

However, this may not reflect the nature of the cytosolic complexes in living bacteria, and their adaptation to external signals. One way to study these properties is fluorescence microscopy using live bacteria. The intrinsic polymerization properties of YscL, N and Q, as detected above and in earlier studies[12,16,48,49], suggest that the formation of dynamic, short-lived complexes in the cytosol is possible. However, no such large complexes have been observed by standard fluorescence microscopy. We hypothesized that detection of such diffusing complexes might require more highly time-resolved microscopy. We thus imaged the fluorescently labelled cytosolic T3SS components YscQ and YscL at different sampling rates, with

**Table 2 | Complexes between the four cytosolic T3SS machinery proteins YscK, YscQ, YscL and YscN can be specifically detected by Halo-tag-based purification.**

| | Non-secreting conditions | | Secreting conditions | |
| --- | --- | --- | --- | --- |
| | **Halo-YscL** | **Halo-YscQ** | **Halo-YscL** | **Halo-YscQ** |
| YscK | ***** | ***** | ***** | ***** |
| YscQ | * | ** | ** | * |
| YscL | ***** | ***** | *** | **** |
| YscN | ** | *** | *** | **** |

LC-MS/MS , liquid chromatography tandem-mass spectrometry; T3SS, type III secretion system; WT, wild type.
Specific interactions between cytosolic T3SS components were identified by LC-MS/MS analysis of eluate samples from pull-down experiments using strains expressing Halo-YscL or Halo-YscQ, respectively. Label-free quantification was used to determine the ratio of interacting proteins in one representative experiment in the strains expressing the respective Halo-tagged proteins and the WT control strains. *, interaction ratio in strain expressing the Halo-tagged protein/WT control >5; **, ratio >20; ***, ratio >80; ****, ratio >320; *****, protein only detected in tagged strains. See Methods section for more details.

exposure times ranging from 400 ms down to 15 ms. For both tested proteins, relatively little distinct moving fluorescence could be detected within the cytosol at 100 and 40 ms exposure time (Supplementary Fig. 8 and Supplementary Movies 1, 2, 4 and 5). At 15 ms exposure, movements of fluorescence within the bacterium are detectable (Supplementary Fig. 8 and Supplementary Movies 3 and 6) and the similarity between different frames, expressed by the Pearson's correlation coefficient, decreases, especially for EGFP-YscL (Supplementary Fig. 8). Such a behaviour would be expected for complexes with a diffusion rate of $1–2 \, \mu m^2 s^{-1}$, as observed for EGFP-YscQ in a recent single-molecule study[39] (see Discussion for details), and thus suggests the presence of unbound cytosolic complexes in live bacteria. This prompted us to study these complexes in more detail using quantitative methods.

FCS is a non-invasive technique that allows the calculation of the apparent diffusion coefficient of fluorophores by measuring the fluorescence intensity in a small focal volume[50]. Fast-diffusing (that is, small) molecules diffuse in and out of the focal volume quickly, leading to fast fluctuations in fluorescence. In contrast, slowly diffusing (that is, large or tethered molecules) fluorophores lead to slow fluctuation. In FCS experiments, the average time a fluorophore spends in the focal volume (the 'transit time') is calculated by fitting the autocorrelation function of fluorescence intensity over time[51,52]. The transit time is inversely linked to the apparent diffusion coefficient of a fluorophore. To validate the existence of cytosolic complexes in live bacteria, we performed fluorescence cross-correlation spectroscopy (FCCS). FCCS allows the detection of codiffusion, and thus interaction, of fluorescently labelled molecules[53]. If two molecules do not codiffuse (that is, they diffuse randomly) through the focal spot, cross-correlation amplitude is '0'. This amplitude increases as the level of interaction increases. To assay the interaction of the two central components of the cytosolic complex, we tested the codiffusion of EGFP-YscQ and mCherry-YscL in a strain lacking the membrane anchor YscD, which, to our knowledge, is the first time FCCS has been applied in bacteria[54]. Despite the relatively high photobleaching rate of mCherry, which led to a lower signal-to-noise ratio in the red channel, we could clearly detect non-zero cross-correlation (thus, codiffusion) between the two proteins, indicative of the formation of soluble complexes between these two T3SS component in live bacteria. In contrast, '0' amplitude was detected in the negative control, where instead of mCherry-YscL, mCherry alone was expressed (Supplementary Fig. 9A).

To study the size and composition of cytosolic complexes in more detail, we performed single colour FCS in live bacteria and determined the diffusion rate of EGFP-labelled soluble T3SS components in strains lacking other members of the cytosolic complex. As a control, we used EGFP, expressed from plasmid at a similar concentration as the EGFP fusions. The diffusion time of EGFP was $1.556 \pm 0.351$ ms (s.d., $n = 39$), equivalent to an apparent EGFP diffusion coefficient of $8.14 \, \mu m^2 s^{-1}$, which fits very well with previously reported values[55–57]. FCS measurements of fusion proteins in the WT strains often showed strong initial photobleaching, indicative of a non-motile, injectisome-bound population of fluorophores in the focus spot. Indeed, we did not observe this phenomenon in the $\Delta yscD$ strains (Supplementary Fig. 10a–c), confirming that all analysed T3SS components are diffuse throughout the cytosol in these strains. Importantly, the diffusion rate of the tested soluble components was comparable in the $\Delta yscD$ and corresponding WT strains (Supplementary Fig. 10d,e). Analysing the $\Delta yscD$ strains prevents any artefacts produced by focusing on a stable injectisome. This allowed us to focus on the cytosolic interactions, and analyse the role of specific players in the T3SS and external cues in the diffusion and composition of the cytosolic complexes.

Our data show that diffusion of the soluble components in the $\Delta yscD$ background significantly differed between different data points for the same protein, both over time and, to an even higher degree, between different bacteria, compatible with a wide range of concurrent complex compositions (Fig. 3a). This suggests that the composition of complexes differs within a bacterial population, possibly caused by a constant turnover of interactions. However, we observed clear differences between the different soluble T3SS components. The two central components of the cytosolic complex, YscQ and YscL, were part of the slowest-diffusing, and hence largest complexes (with apparent diffusion coefficients of 2.41 and $2.06 \, \mu m^2 s^{-1}$ under non-secreting conditions, respectively), while the peripheral components, YscK and YscN, mostly diffuse faster (apparent diffusion coefficients of 2.86 and $3.27 \, \mu m^2 s^{-1}$) (Fig. 3a).

To characterise the soluble cytoplasmic T3SS and the role individual protein play in forming these complexes, we deleted single T3SS components in the labelled strains. In all of the tested cases, we observed a significant increase in diffusion, indicating decreased complex size (Fig. 3b). This strongly suggests that all soluble components are part of the complexes. However, different deletions had distinct effects, suggesting that complex formation is a gradual, rather than an all-or-nothing process. For example, while a deletion of YscN significantly impacted the size of YscQ-containing complexes, it did not decrease the complex size as strongly as the point mutation $YscQ_{M218A}$, which prevents internal translation initiation in the C-ring protein and hence expression of $YscQ_C$[17,18], a mutation shown to abolish oligomerization of the C-ring protein[48] and hence C-ring formation[39] and secretion[18]. Similarly, YscL complexes were more strongly affected by deletion of YscQ than YscN. In the cases where the only proposed direct interaction partner was missing (EGFP-YscK $\Delta yscQ$ and EGFP-YscN $\Delta yscL$), the diffusion time was slightly higher than for EGFP alone, and

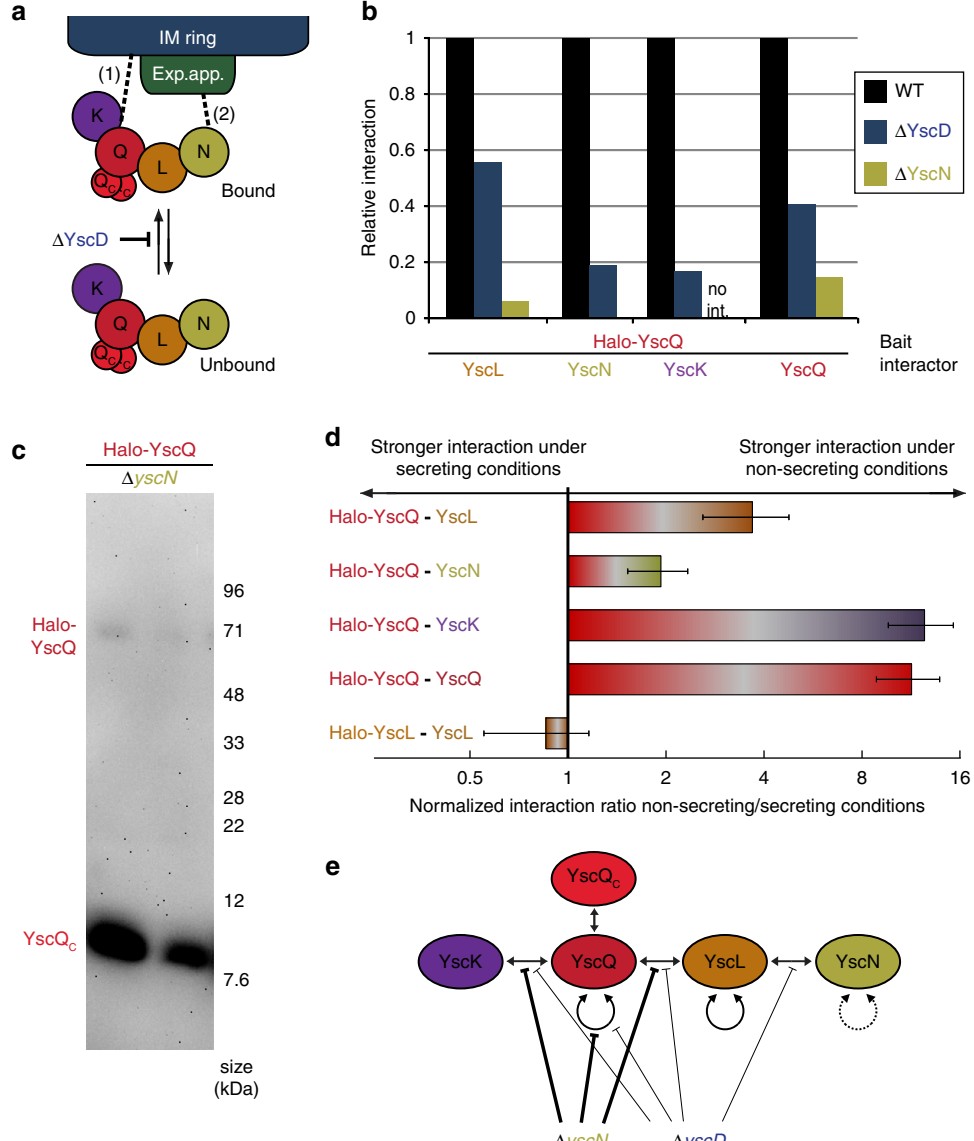

**Figure 2 | Stable interactions between the soluble components occur within the cytosol. (a)** Schematic depiction of the bound (top) and unbound (bottom) cytosolic complex, colours and full protein names as in **e**. IM ring is depicted in blue, and export apparatus in green. Presumed connections between the cytosolic complex and the membrane-bound part of the injectisome indicated by dashed lines: (1) YscK or YscQ to IM ring, (2) YscN to export apparatus, possibly via YscO. The absence of YscD prevents binding of the cytosolic complex to the basal body. **(b)** Relative amounts of interaction (as detected in one representative quantitative mass spectrometry experiment, normalized to WT) between the given proteins in strains otherwise WT (black), or strains lacking YscD (blue) or YscN (dark yellow). No int., no interaction detected (YscQ:YscK in strain lacking YscN). **(c)** Interactions of Halo-YscQ (expected molecular weight 69.5 kDa) and $YscQ_C$ (expected molecular weight 10.0 kDa), detected in the purification eluate by immunoblot using a polyclonal anti-YscQ antibody. **(d)** Ratio of amounts of interaction (as detected in one representative quantitative mass spectrometry experiment) between non-secreting and secreting conditions for the given proteins (first protein Halo-tagged bait, second protein detected and quantified in eluate by mass spectrometry), corrected for the increase in protein levels under secreting conditions (error bars represent the s.d. in expression levels, as determined by the fluorescence intensity of 60 spots per strain, see Supplementary Table 3 for details). **(e)** Schematic representation of the influence of YscD and YscN on the network of stable interactions within the cytosolic T3SS components. There is conflicting evidence on a possible direct interaction of YscN and YscQ, which is not included in this scheme[22,23,72]. Dashed line (YscN:YscN interaction), not tested.

similar to that expected for a monomer of this size (Fig. 3b). In contrast, EGFP-YscL $\Delta yscQ$ showed a slightly slower diffusion, suggesting dimer formation[15,58].

Interestingly, the apparent diffusion coefficients of the cytosolic complexes in the absence of the IM ring changed upon activation of the system by chelating $Ca^{2+}$ in the surrounding medium. The diffusion rates of YscQ, YscL and YscK increased under secreting conditions (Fig. 3c). This effect was most pronounced for YscK, where the diffusion rate increased from 2.86 to 4.17 $\mu m^2 s^{-1}$.

In contrast, the diffusion of cytosolic GFP did not change with extracellular $Ca^{2+}$ levels (transit time of 1.556 ± 0.351 ms under non-secreting conditions versus 1.564 ± 0.447 ms under secreting conditions, $P = 0.749$ in a two-tailed homoscedastic $t$-test).

To analyse this unexpected effect and its connection to effector secretion in more detail, we performed FCS experiments on the two central soluble T3SS components YscQ and YscL at different intermediate calcium levels, and correlated the results with the secretion of effectors in the WT strain under the same conditions.

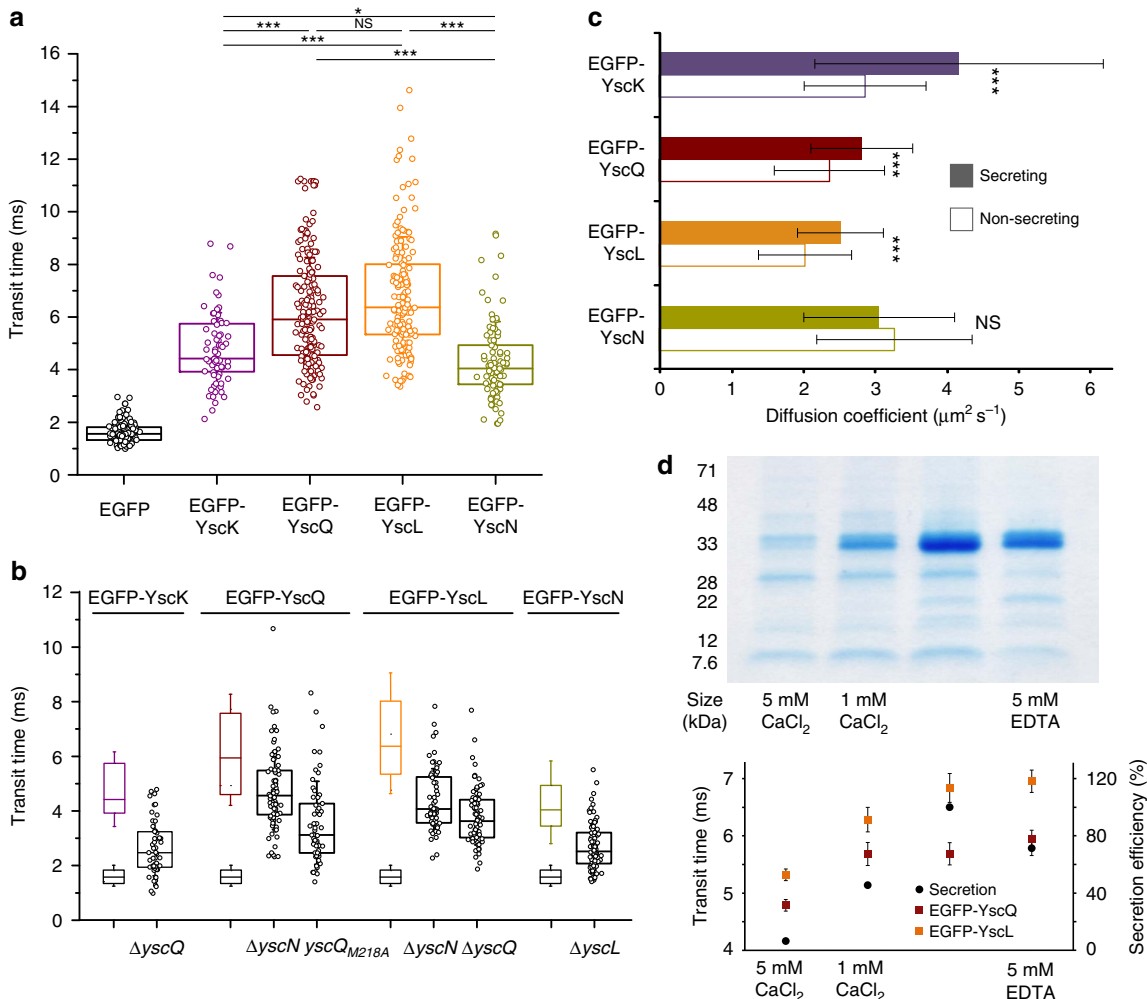

**Figure 3 | The soluble T3SS components form a flexible and adaptive network of interactions in live bacteria. (a)** Diffusion times of complexes including the soluble N-terminally EGFP-tagged components YscK, YscQ, YscL or YscN in strains lacking YscD, under non-secreting conditions (control: EGFP expressed from plasmid at a similar expression level). Box plots display single data points (average transit time over 5 s) as well as 25th, 50th and 75th percentiles; whiskers indicate s.d. $N = 73$–199 apparent diffusion coefficients from at least 30 different bacteria in two to six independent experiments per strain. **(b)** Deletions of other cytosolic T3SS machinery proteins differentially affect the size of the cytosolic complexes. Effect of deletions of single other members of the cytosolic complex on the transit time of complexes including N-terminally EGFP-tagged components. On the left side of each panel, the transit times of the corresponding $\Delta yscD$ strain (top, coloured) and EGFP (bottom, black) are indicated for comparison. $N = 54$–86 apparent diffusion coefficients from at least 25 different bacteria in two to four independent experiments per strain. **(c)** Diffusion of cytosolic complexes including YscK, YscQ or YscL increases upon activation of the T3SS by chelation of $Ca^{2+}$. Apparent diffusion coefficients of N-terminally EGFP-tagged components in strains lacking YscD under secreting (filled bars) or non-secreting (open bars) conditions. Error bars represent s.d.'s from 73 to 158 average apparent diffusion coefficients from at least 30 different bacteria in two to four independent experiments per strain and condition. To ensure the comparability of the different conditions, only data from experiments where both conditions were directly compared is included in this figure. **(d)** Top, representative secretion profile showing the secreted proteins in a WT strain at different calcium/EDTA levels as indicated. Bottom, correlation between the transit time of EGFP-YscQ (dark red) and EGFP-YscL (orange) in strains expressing these proteins in absence of YscD (left axis, $n = 64$–158), and the secretion rate in the WT strain as displayed in the top part ($n = 3$, normalized to the maximal rate, right axis), both measured under the same conditions. Error bars represent s.e.m. value; a direct correlation and statistical analysis can be found in Supplementary Fig. 11. *$P < 0.05$; ***$P < 0.001$; NS, no statistically significant difference in a two-tailed homoscedastic $t$-test.

Our results show that across the range 5 mM CaCl$_2$ to 5 mM EDTA, the diffusion of the studied T3SS components largely parallels the relative secretion efficiency in the WT strain under the same conditions (Fig. 3d and Supplementary Fig. S11). Interestingly, the addition of 5 mM EDTA (fourth data point in Fig. 3d) did not further enhance effector secretion, but did slow down diffusion when compared to addition of neither CaCl$_2$ nor EDTA (third data point).

To uncover the molecular basis for this change in diffusion of the cytosolic complexes, we performed interaction experiments using the strains expressing Halo-tagged cytosolic T3SS components used in Fig. 2a–c under secreting and non-secreting conditions, quantified the protein interactions with quantitative mass spectrometry and corrected the resulting interactions, taking into account the different expression levels of the cytosolic T3SS components under these conditions (Supplementary Table 3 and see Methods for details). The changes in the stable interactions (Fig. 2d) largely mirror the results obtained in our FCS experiments: the interaction between YscQ and YscK, as well as homo-oligomerization of YscQ, was significantly stronger under

non-secreting conditions. The interaction between YscQ and YscL showed an intermediary phenotype. In contrast, the interaction between YscQ and YscN was only slightly affected, and the homo-oligomerization (mostly like dimerization) of YscL remained almost constant.

## Discussion

In this study, we analysed the interactions and dynamics of the soluble T3SS components that play a crucial, but so far little understood, role in the export of proteins by the T3SS. Combining live fluorescence microscopy, a gentle and specific co-immunoprecipitation method and fluorescence correlation spectroscopy to analyse directly the diffusion of proteins in live bacteria, we confirmed the assumption that the soluble components form a large complex at the proximal interface of the injectisome. This complex involves and requires all four soluble T3SS components, the ATPase SctN/YscN, its negative regulator SctL/YscL, the C-ring protein SctQ/YscQ and the accessory protein SctK/YscK. In addition, these proteins also form dynamic diffusing cytoplasmic complexes that are not bound to the basal body. Intriguingly, the diffusion of these complexes was influenced by the external conditions governing effector secretion, suggesting a previously unidentified mechanism for bacteria to regulate type III secretion.

Using conventional fluorescence microscopy, we found that each of the soluble components localize to the injectisome, requiring the presence of all other partners (Fig. 1b–d). Similar results had been obtained for SctQ/YscQ, SctN/YscN and the flagellar ATPase FliI[29,59]. The consistent colocalization of all the components of the cytosolic complex with the IM ring component SctD/YscD (Fig. 1c) indicates that there are few basal bodies without attached cytosolic complexes containing all soluble components. More quantitatively, this notion is supported by the similar number of spots per bacterium for all T3SS components (Table 1). However, detailed analysis showed that the number of spots per bacterium was slightly (7–20%) lower for the soluble components, compared to the IM ring protein SctD/YscD, which raises the possibility that not all components are present in all injectisomes. In support of this hypothesis, a recent structural study found a lower electron density in the cytosolic complex of a subset of injectisomes[38].

Using fluorescent fusion proteins for all soluble T3SS components, we could estimate the relative stoichiometry of the injectisome-bound cytosolic complex by comparing the intensity of fluorescent injectisome foci in a three-dimensional analysis. Based on the well-established stoichiometry of SctD/YscD, assumed to be to present in 24 copies[30,42–45] as a standard, we observed values close to the previously published values for SctV/YscV[60] and SctQ/YscQ[39]. SctL/YscL was present in ∼14 copies, roughly compatible with a 2:1 ratio of SctL/YscL:SctN/YscN for a SctN/YscN hexamer. Surprisingly, SctN/YscN itself was present in ∼12 copies, which is at odds with the expected presence as a hexamer. While the high cytosolic fluorescence and possible partial degradation for SctN/YscN (Fig. 1f and Supplementary Fig. 1b) might impact the quantification of spot intensity for this protein, it is intriguing to note that dodecamers of SctN/YscN have been observed by cryoelectron microscopy[61] and were even found to be the predominant and most active form of the T3SS ATPase at the membrane[12]. Further, a recent fluorescence-based investigation of the flagellar ATPase FliI also found an average number of more than six FliI at the flagellar motor, and the authors proposed a coexistence of a FliI hexamer as a static substrate loader and FliH$_2$–FliI complexes as dynamic substrate carriers[62]. Finally, the 'accessory protein' SctK/YscK was present in around six copies, which is compatible with a role in

linking each of the six 'pods' of the cytosolic complex to the IM ring.

Analysing the fluorescence distribution across bacteria in more detail, we found that the soluble components display a substantial amount of fluorescence in the cytosol. This was not the case for the two tested membrane-bound components where the proteins were primarily localized to the injectisome spots (Fig. 1f). To study the interactions in the cytosol as well as at the injectisome, we used a HaloTag-based co-immunoprecipitation protocol. Using the central components SctQ/YscQ and SctL/YscL as bait, we found that all soluble T3SS components are able to form stable interactions in the cytosol (Table 2 and Fig. 2). This is in agreement with the results of Johnson and Blocker[47] in Shigella flexneri, based on co-immunoprecipitation, but differs from the sorting platform identified by Lara-Tejero et al.[20] in Salmonella typhimurium, which does not include the ATPase and did not show any dependence on the presence of the ATPase for its interaction. The interaction strength was significantly reduced in the absence of SctD/YscD (Fig. 2b), indicating that the complexes in the cytosol are less stable than those injectisome-bound. Given that the proximal complexes are thought to be stabilized by a twofold connection with the basal body (Fig. 2a), as is the case in the flagellum[36,63], possibly through SctK/YscK (to the IM ring) and SctN/YscN (to the export apparatus via SctO/YscO), such a behaviour might be expected. Our finding that interactions of the peripheral components of the cytosolic complex, SctK/YscK and SctN/YscN, are most affected by the absence of SctD/YscD, also agrees with this hypothesis.

Deletion of a member of the cytosolic complex itself, the ATPase SctN/YscN, had a much more drastic effect on the complex than deletion of SctD/YscD (Fig. 2). While in both cases, the soluble components are completely cytosolic (Fig. 1d)[29], SctN/YscN seems to be required for all interactions within the cytosolic complex that are stable enough to be detected by the applied co-immunoprecipitation method.

To detect less stable complexes and understand their dynamics in live bacteria, we performed live fluorescence microscopy at different frame rates. Our results indicated the presence of unbound cytosolic complexes that were only detectable at very short (15 ms) exposures (Supplementary Fig. 8). Such a behaviour would be expected for complexes with a diffusion rate in the range detected in our earlier single molecule studies of SctQ/YscQ[39] and this study. At a diffusion rate of 1.5 μm$^2$ s$^{-1}$, the two-dimensional mean-squared displacement within exposure times of 100, 40 and 15 ms is 0.6, 0.24 and 0.09 μm$^2$, respectively, compatible with a detection of a diffusing complex only in the latter case (resulting in an average displacement of 0.3 μm within the exposure time).

We thus analysed the interactions of the cytosolic components inside live bacteria in more detail, using fluorescence (cross-) correlation spectroscopy (FCS/FCCS). FCS and FCCS confirmed the presence of soluble complexes in the cytosol (Fig. 3a and Supplementary Fig. 9). Additionally, it yielded a much more multifaceted image of the interaction network. The detected complexes had widely varying apparent diffusion coefficients, indicative of significantly different sizes, and hence a dynamic and adaptive interaction network. This suggests that there is not one clearly defined 'cytosolic complex' in live bacteria, but that new components can be incorporated at any point, probably as monomers, as we did not observe clear steps in diffusion.

In theory, it should be possible to deduce the size of diffusing complexes from the diffusion coefficient, based on the Stokes–Einstein equation. However, it has become evident that diffusion of proteins within the bacterial cytosol is far from the theoretical (diffusion coefficient)–(molecular weight)$^{-1/3}$ correlation, as already indicated in ref. 55 and shown more

systematically in ref. 57. In addition, despite the limited height of bacteria, fluctuations of fluorescence intensity in the z-direction may occur that are not accounted for by our FCS analysis. While we cannot directly deduce the size of the cytosolic complexes, we hypothesize that for the same protein, faster diffusion indicates a smaller complex. Analysing the diffusion of the soluble T3SS components in strains lacking other soluble components indicates whether the missing protein is part of the original complex and how the studied complex is influenced by its absence. Strikingly, all tested deletions had significant, but different, impacts on the detected complexes (Fig. 3b). The finding that incomplete cytosolic complexes can form in the absence of different member components (Sct/YscK, L, Q, L and N) is interesting, given the absolute necessity of all cytosolic T3SS proteins for the assembly of other cytosolic proteins at the injectisome. This suggests that while incomplete cytosolic complexes are able to form (Fig. 3), they are not stable enough (Fig. 2) for proper assembly at the injectisome (Fig. 1). This is particularly evident for the ATPase, which has a relatively mild influence on the formation of diffusing soluble complexes in live cells, but has an important role in stabilizing these complexes, as highlighted by the strong decrease of interactions in our co-immunoprecipitation in the absence of SctN/YscN.

Significantly, the complexes involving SctL, the negative regulator of the ATPase, are more strongly affected by deletion of the C-ring component SctQ/YscQ than the ATPase SctN/YscN itself, suggesting that SctL/YscL can still form larger complexes with SctQ/YscQ. However, in the absence of SctQ/YscQ, it probably forms a smaller complex with the ATPase, similar to the previously shown $SctN–SctL_2$ complex[16,62]. Besides the $SctN–SctL_2$ complex, another complex involving the soluble T3SS components has been described in Salmonella SPI-1, a sorting platform involving SctK, Q and L[20]. Given that SctL participates in both of these complexes, we expected an increase in the formation of the sorting platform upon deletion of SctN. However, in the absence of SctN, the interaction between SctL/YscL and SctQ/YscQ was markedly decreased (Fig. 2b), suggesting that there is no competition between these complexes.

The peripheral components of the unbound cytosolic complex, SctK/YscK and SctN/YscN (see Fig. 2a), appear to be bound least tightly. This is indicated both by a stronger decrease in stable interactions in the $\Delta yscD$ strain for these two proteins (Fig. 2b) and the strong effect of deletions of their direct binding partners, SctQ/YscQ and SctL/YscL, respectively, in FCS (Fig. 3b). This is in line with the findings of Johnson and Blocker[47] for SctK, and might be an explanation for the absence of the ATPase in the SPI-1 sorting platform[20]. The affinity of SctN for the SctKQL complex might be low enough in Salmonella to not be detected by blue native polyacrylamide gel electrophoresis (PAGE), and have little influence on the stability of the SctKQL complex. In contrast, our study shows that in Yersinia, the ATPase is required for stable interactions between these proteins, and in turn interacts with the sorting platform components (Fig. 2).

Unexpectedly, the stability of the cytosolic interactions did not only depend on the available proteins but also on the environmental conditions, as indicated by the influence of the external $Ca^{2+}$ levels on the diffusion (Fig. 3d). This faster diffusion under secreting conditions is reminiscent of the higher exchange rate of the C-ring under secreting conditions[39] and suggests that both the injectisome-bound and -unbound cytosolic complex might rearrange upon activation of secretion. This speculation concurs with the recent findings of Nans et al.[37], who detected a change in the structure of the cytosolic complex in cryoelectron tomograms of Chlamydia trachomatis T3SS upon activation of the system. Notably, the direct influence of external

$Ca^{2+}$ levels on the cytosolic complex, even in strains lacking T3SS needles (as is the case in the studied $\Delta yscD$ strains), represents an additional way for the T3SS to respond to external cues, in parallel to the release of the internal gatekeeper protein by host cell contact, which is signalled from the needle tip through rearrangement of the needle[32–34]. This approach may ensure beneficial interaction of the soluble components with both the export cargo and the injectisome, leading to more specific and faster export of the correct cargo, an important benefit in the interaction with host cells.

To determine the cause for the influence of external calcium on the soluble T3SS components, we compared the network of stable interactions, as determined by quantitative mass spectrometry of the interactors of the Halo-labelled soluble T3SS components, under secreting conditions (Fig. 2d). The surprisingly strong decrease of some interactions, especially YscQ-YscQ and YscQ-YscK, provides an intriguing possible explanation for this effect: changing extracellular calcium levels lead to changes in protein affinities inside the bacterium.

How the external $Ca^{2+}$ concentration is sensed by the soluble components remains unclear. One possibility is that extracellular calcium is, at least to some degree, taken up by the bacteria. Indeed, the presence of a T3SS lead to stronger $Ca^{2+}$ binding at low external concentrations in Yersinia pestis[64], and a recent publication showed that the interaction of SctP/EscP and SctW/SepL, proteins involved in needle length determination and substrate control of the enteropathogenic E. coli T3SS, also is influenced by the extracellular calcium concentration[65]. Another possibility is that SctK/YscK, which interacts with the bacterial membrane even in the absence of other T3SS components[21], senses the extracellular calcium level, directly or indirectly, by an unknown mechanism. In line with this hypothesis, the difference between the size of the soluble cytosolic complexes in the $\Delta yscD$ background under secreting and non-secreting conditions was most prominent for SctK/YscK (Fig. 3c), and the interaction between SctK/YscK and SctQ/YscQ was most strongly affected under secreting conditions (Fig. 2d). Taking into account the previously observed chain of interactions of the soluble T3SS components, our results suggest that SctK/YscK might connect the C-ring and the IM ring, accounting for the additional electron density recently observed by Hu et al.[36]. Upon induction of secretion by chelation of $Ca^{2+}$, the cytosolic interaction between YscK and YscQ would be weakened. This might lead to the increased exchange of YscQ between the injectisome and the cytosolic pool observed earlier[39], and in turn allow effector export by the T3SS. Such a mechanism would be unique to the T3SS and not be required in the flagellum. Intriguingly, SctK/YscK, which we propose to act as sensor or transducer of this signal, is the only soluble T3SS component that does not have a flagellar homologue.

In summary, our data provide insight into the dynamic and adaptive interaction network of the soluble T3SS in the cytosol and its relation with the bound cytosolic complex at the injectisome. Based on these results, we propose a mechanism by which changing external conditions can govern protein export by the T3SS injectisome, and suggest a role for the T3SS-specific protein SctK/YscK.

## Methods

**Bacterial strains and genetic constructions.** All strains and plasmids used in this study are listed in Supplementary Table 2. All fusion proteins used in this study were stably introduced by exchange of the WT gene in the pYV virulence plasmid by two-step homologous recombination[40] and are hence expressed from their native genetic environment. Plasmids were generated using Phusion polymerase (Finnzymes, Espoo, Finland). Mutators for modification or deletion of genes were constructed as described in ref. 41. All constructs were confirmed by sequencing (Source BioScience, Oxford, UK).

*E. coli* Top10 and BW19610 were used for cloning *and E. coli* SM10 λ pir $^+$ for conjugation (all strains were provided by Guy R. Cornelis, Basel, Switzerland). *E. coli* were grown on Luria–Bertani agar plates or in liquid Luria–Bertani medium at 37 °C. To select for expression vectors and suicide vectors, ampicillin (200 µg ml$^{-1}$) or streptomycin (100 µg ml$^{-1}$) were added to the cultures and plates. All *Y. enterocolitica* strains used in this study are derivates of the strain IML421asd (ref. 66), which lacks the main virulence effectors YopH, YopO, YopP, YopE, YopM and YopT, and is additionally auxotrophic for diaminopimelic acid due to a mutation in the aspartate-β-semialdehyde dehydrogenase (*asd*) gene. *Y. enterocolitica* were routinely grown at 28 °C in brain heart infusion (BHI) broth containing nalidixic acid (35 µg ml$^{-1}$) and diaminopimelic acid (80 µg ml$^{-1}$).

**Y. enterocolitica cultures for secretion and microscopy analysis.** Cultures for protein secretion assays and further microscopy analysis were inoculated from stationary overnight cultures (a) to an optical density at 600 nm (OD$_{600}$) of 0.15 in BHI broth containing 5 mM EDTA (secreting conditions) or (b) to an OD$_{600}$ of 0.12 in BHI broth containing 5 mM CaCl$_2$ and filtered through a 0.45 µm syringe-top filter (non-secreting conditions). In both cases, cultures were supplemented with nalidixic acid (35 µg ml$^{-1}$), diaminopimelic acid (80 µg ml$^{-1}$), glycerol (4 mg ml$^{-1}$) and MgCl$_2$ (20 mM). After 1.5 h of growth at 28 °C, the *yop* regulon was induced by shifting the culture to 37 °C in a shaking water bath to ensure fast temperature shift. After 3 h of agitation at 37 °C, the optical density at 600 nm of the culture was recorded, and cultures were used for further analysis.

To determine the secretion efficiency at different external Ca$^{2+}$ levels (Fig. 3d), 1 ml of non-secreting culture was collected at this point (2,400g, 4 min), and gently resuspended in 1.5 ml of imaging buffer (M22 buffered with 20 mM HEPES instead of phosphate, 80 µg ml$^{-1}$ diaminopimelic acid, 0.4% glycerol, 20 mM MgCl$_2$, 0.4% casamino acids) supplemented with the given concentrations of CaCl$_2$ or EDTA. The resuspended bacteria were agitated at 37 °C for 2 h and then further processed as described below.

Bacterial total cell and supernatant fractions for immunoblot analysis and secretion profiles, respectively, were separated by centrifugation (5–10 min, 18,000–21,000g). Unless mentioned otherwise, proteins secreted by $3 \times 10^8$ bacteria or produced in a cell pellet of $2.5 \times 10^8$ bacteria were loaded per lane. Secreted proteins were precipitated with a final concentration of 10% trichloroacetic acid for 1–8 h at 4 °C. Proteins were separated on Novex 4–20% gradient SDS–PAGE gels (Life technologies). Secreted proteins were stained using the Coomassie-based 'Instant blue' staining solution (Expedeon), while immunoblotting was carried out using mouse polyclonal antibodies against GFP (Clontech 632459; 1:1,000) or rabbit polyclonal antibodies against YscQ (MIPA235; 1:1,000).

**Fluorescence microscopy.** For fluorescence microscopy analysis, *Y. enterocolitica* cultures were used after 3h incubation at 37 °C (see above). Eight hundred microlitres of cultures were pelleted by centrifugation (2,400g, 4 min) and resuspended in 400 µl (for non-secreting conditions)/300 µl (for secreting conditions) of imaging buffer (M22 buffered with 20 mM HEPES instead of phosphate, supplemented with 80 µg ml$^{-1}$ diaminopimelic acid, 0.4% glycerol, 20 mM MgCl$_2$, 0.4% casamino acids) including 5 mM CaCl$_2$ for non-secreting conditions or 5 mM EDTA for secreting conditions, unless stated differently.

Resuspended bacterial culture of 1.5 µl was placed on a microscope slide layered with a pad of 2% agarose in imaging buffer. Images were taken on a Deltavision Spectris Optical Sectioning Microscope (Applied Precision, Issaquah, WA, USA), equipped with a UPlanSApo × 100/1.40 oil objective (Olympus, Tokyo, Japan) and × 1.6 auxiliary magnification, using an Evolve EMCCD Camera (Photometrics, Tucson, AZ, USA) at a gain level 50. Unless stated differently, exposure times were 5 ms for differential interference contrast images, 400 ms for mCherry fluorescence, using an mCherry filter set, and 200 ms for GFP fluorescence, using a GFP filter set. In dual colour imaging experiments, mCherry fluorescence was excited and recorded before GFP fluorescence to minimize photobleaching of mCherry. Per image, a z stack containing 7 to 15 frames per wavelength with a spacing of 150 nm was acquired.

**Data processing of fluorescence microscopy.** The z stacks were deconvolved using SoftWoRx 5.5 (Applied Precision, Issaquah, WA, USA) with the conservative ratio settings and manual background determination. Deconvolution is an image processing technique to improve the contrast and resolution of fluorescence microscopy images by reassigning the blurring present in microscope images induced by the limited aperture of the objective. Representative differential interference contrast and fluorescence frames were selected and further processed with the ImageJ software[67]. When comparing different labelled proteins, the same visualization settings were used for each image to ensure comparability. Line profiles were obtained manually using ImageJ; 32 line profiles from six different fluorescence images were normalized and combined for the representation. Spots were detected using the Imaris software package (Bitplane) using the following settings: spot *xy* diameter = 0.24 µm, *z* diameter = 0.50 µm, background subtraction enabled, quality > 400, distance from *xy* border > 0.4 µm and distance from *z* border > 0.1 µm. The unspecific cytosolic background was determined as average fluorescence intensity within unlabelled bacteria, as shown in Supplementary Fig. 5c. The area for the background determination was

determined by the shapes algorithm in Imaris, using the following settings: smoothing enabled, surface grain size: 0.4 µm, local background enabled, diameter of largest sphere fitting into object: 0.8 µm, threshold = 400 and voxels > 100. The mean background-corrected intensity of 1,900–3,640 detected spots per strain was calculated to compare the respective fluorescence intensities for the relative stoichiometry of components.

To compare expression levels of the fusion proteins under secreting and non-secreting conditions, we used a simplified version of this protocol: the maximal intensity of 60 spots from three deconvoluted fields of view per strain was corrected for the average background fluorescence in 30 unlabelled bacteria. The ratio of spot intensity under secreting and non-secreting conditions was normalized using a ratio of 2.45 for YscQ[68] as a reference.

**HaloTag-based interaction studies, mass spectrometry and quantification.** Bacterial cultures were incubated under non-secreting conditions, unless specified otherwise, as described above. A negative control was included with each pull-down experiment, using a strain expressing no HaloTag fusion protein (untagged WT). Cells were pelleted (10 min, 6,700g) and resuspended in HaloTag protein purification buffer (Promega). Twenty millimoles of MgCl$_2$ was added to the buffer to retain YscN ATPase integrity. Cells were lysed by freeze-thaw, treated with 0.08 mg ml$^{-1}$ lysozyme for 30 min at ambient temperature and then sonicated to clarification. The lysate was pelleted by centrifugation (30 min, 10,000g, 4 °C). Magne HaloTag beads (Promega) were prepared in parallel according to the manufacturer's protocol. The beads were then incubated with the lysate supernatant with mixing for 2 h at 4 °C to allow linkage of the HaloTag to the beads. The beads were subsequently washed for 5 min with HaloTag purification buffer and resuspended in 1 × SDS loading buffer according to the final OD$_{600}$ of the initial cultures. Interacting partners of the HaloTag fusion proteins were eluted by heating to 95 °C for 2 min.

Western blotting and mass spectrometry were used for characterization of the sample eluates. For mass spectrometry, samples were run 5–10 mm on a reducing 4–20% SDS–PAGE gel and excised. Excised gel fragments were prepared and digested using sequencing-grade trypsin (Promega) according to a standard protocol used by the Oxford University Central Proteomics Facility. All samples were analysed in the Oxford University Central Proteomics Facility using a Thermo Q Exactive Orbitrap LC-MS/MS Mass Spectrometer. Label-free quantification was performed using the SINQ tool from the Central Proteomics Facility Pipeline[69]. Table 2 indicates the enrichment factor of detected interacting proteins in the strains expressing Halo-YscL or Halo-YscQ, over the untagged WT control strains, as determined by the ratio of their spectral indices, a measure of protein representation in each sample. Representative results from one large experiment comparing all strains in parallel are shown. Where possible, these results were confirmed by immunoblot analysis, yielding consistent results.

**Fluorescence correlation and cross-correlation spectroscopy.** Cultures for FCS and FCCS experiments were grown under non-secreting conditions. Where applicable, secretion was induced by resuspending in imaging buffer containing EDTA instead of CaCl$_2$ (see above) and layering the bacteria on a 2% agarose patch resuspended in the same buffer. One hundred and fifty microlitres of agarose solution were pipetted into GeneFrame adhesive chambers (Thermo Fisher), and 1.5 µl of bacterial resuspension were layered on top of the agarose pad, which was then sealed with a large cover slip.

FCS was carried out with a Zeiss 780 and Zeiss 880 Microscope equipped with a spectral GaAsP detector. All the measurements were done with a × 40, NA 1.2, water-immersion objective (FCS objective) with a correction collar. The correction collar was aligned before every measurement to avoid aberrations due to the refractive index mismatch. GFP-labelled molecules were excited with 488 nm laser and emission was collected between 505 and 550 nm. Laser power of 2–5 µW was used to avoid photobleaching. For FCCS measurements, GFP-labelled molecules were treated as above, mCherry-labelled molecules were excited using a 561 nm laser, and emission was collected at 615 nm and above to avoid bleedthrough of the GFP signal into the red channel.

The laser light was focused on the labelled bacterium and the count rate per molecule was maximized. After focusing, three repetitions of 5 s measurements were obtained. Only one series of three measurements was carried out on one bacterium. Obtained FCS curves were fitted with two-dimentional + triplet model to obtain the transit diffusion time and the number of particles, $G(\tau) = \frac{1}{N}(1 + \frac{\tau}{\tau_D})^{-1}[1 + T(1 - T)^{-1}\exp(\frac{-\tau}{\tau_{Tr}})]$, using the FoCuS-point software[70]. Each fit was evaluated individually; in case the fit quality was not good enough (that is, when $\chi^2 > 0.05$), the according FCS data was excluded.

**Statistical analysis.** Unless indicated otherwise, all displayed centre values represent the mean (average) value, and error bars represent the s.d. of the value. *P* values were calculated using two-tailed equal variance *t*-tests. If not explicitly stated in the respective figure legend, the exact sample sizes for each experiment are listed in Supplementary Table 4.

**Data availability.** All relevant data are available from the authors. Uncropped versions of the displayed gels and immunoblots are provided in Supplementary Fig. 12.

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

## Acknowledgements

C.E. and E.S. were supported by the Wolfson Foundation, the Medical Research Council (MRC) (Grant no. MC_UU_12010/Unit Programmes G0902418 and MC_UU_12025), MRC/BBSRC/ESPRC (Grant no. MR/K01577X/1) and the Wellcome Trust (Grant ref. 104924/14/Z/14). E.S. was supported by EMBO Long Term and Marie Curie Intra-European Fellowships (MEMBRANE DYNAMICS). This work was supported by a Wellcome Trust (http://www.wellcome.ac.uk/) Strategic Award (091911) supporting advanced microscopy at Micron Oxford. We would further like to acknowledge Christoffer Lagerholm, Esther Garcia and Dominic Waithe and generally the Wolfson Imaging Centre Oxford for microscope access and technical help during the project.

## Author contributions

A.D. and J.P.A. designed experiments, interpreted data and wrote the manuscript. A.D., E.S., M.H. and T.M. performed experiments. All authors discussed the results and commented on the manuscript.

## Additional information

**Competing interests:** The authors declare no competing financial interests.

