## [Peer Review File · Nature Communications]

Reviewers' comments:

Reviewer #1 (Remarks to the Author):

In this study, the authors utilized some cutting edge technologies to demonstrate the interactions among the T3SS cytosolic proteins and differences in the assembly under inducing and non-inducing conditions, providing insights into the dynamic and adaptive interaction network of the soluble T3SS in the cytosol and its relation with the bound cytosolic complex at the injectisome. However, previous studies in *Yersinia*, *Salmonella* and *Shigella* on similar topics, addressed by conventional methods of protein-protein interaction, predicted most of the outcomes of the current study, thus the results are largely confirmatory in nature. The authors did make an important new observation, a type III needle independent sensing of the calcium for the intracellular complex formation, suggesting a previously unidentified mechanism for the regulation of type III secretion. Unfortunately, this was not pursued further in the current study.

Minor points:

1. Fig. 2C, the protein levels in bacterial lysate and a negative control should be included. The size differences between YscQ and YscQc are not clear from the gel.
2. Line 27, "..... regulation type III secretion.", missing "of".
3. Line 37, it is *exoU* gene, not the "functional T3SS", that is associated with increased antibiotic resistance.

Reviewer #2 (Remarks to the Author):

The authors studied the four soluble proteins involved in the injectisome. In this manuscript that is clearly written, the protein localisations are observed upon deletion of several proteins involved, performing co-immunoprecipitation experiments and FCS. Fluorescence microscopy showed that each of the soluble components localize to the injectisome, requiring the presence of all other partners and intensity analysis of the foci resulted in estimates of the relative stoichiometry of the injectisome complex. Coimmunoprecipitation revealed that all interactions were present, but significantly reduced in absence of YscD, indicating that the complexes in the cytosol are less stable than the injectisome-bound ones.

The microscopy and immunoprecipitation studies seem to be solid but I have some questions concerning the fluorescence fluctuation studies:

The authors tried to study the interaction of the soluble injectosome proteins using FCCS. Figure 3A shows the auto- and cross-correlation curves of a negative control and a YscQ and YscL sample. The negative sample clearly gives a near-zero amplitude as anticipated and the Ysc-sample a clear positive amplitude. However, as the authors mention in the main text the tagged mCherry protein is sensitive to photobleaching. A real danger in FCCS experiments is when one of the signals decreases strongly (e.g. due to photobleaching) and the other fluorescence signal decreases as well, maybe only slightly (minor bleaching). In this case a strong false-positive cross-correlation amplitude could be observed that could easily overwhelm a true low-amplitude cross-correlation signal. In Figure 1A-A it's now difficult to judge the quality of the cross-correlation signal since this blue graph has been plotted behind the autocorrelation curves. In order to see if the cross-correlation adds reliable info to the manuscript I would like to see the two original intensity traces and a different way of plotting the blue graph, such that the whole curve can be seen.

In addition, I question why the authors did not use FCCS to study all the interactions in the soluble injectosome complex because this could give a wealth of information about the composition of the complexes. The authors claim they are the first ones measuring FCCS in bacteria but I still doubt if these kind of measurements are possible in a reliable manner (see remark above) due to the

limited amount of labeled protein, the small confined bacterial volume especially in combination with the problems of photobleaching of the photo-unstable yellow/orange/red fluorescent proteins. Is this also the reason the authors continue their analysis in the manuscript using only single colour FCS (using the more photostable eGFP)? As such the FCCS experiment shown here does not add much to the story.

The authors use the diffusion time for a qualitative but also quantitative comparison between different strains. Although the qualitative comparison probably holds I have some doubts about the absolute diffusion coefficients presented in the article. At page 22 the authors state they have used a 2-dimensional diffusion plus triplet model for fitting the autocorrelation curve. The equation of the full model or a reference should be included in the text. In addition, I wonder how accurate the obtained diffusion coefficients are since due to the limited height of the bacteria, intensity fluctuations and diffusion in the z-direction of the inhomogenous detection volume is most likely severely constrained but certainly occurring. I guess in this case it would be better to use a constrained diffusion model with a limited z-profile (like presented by Generich et al (BiophysJ) and others) for fitting? It's probably a good idea to include a typical correlation curve including fit and fit-residuals in the supplementary info as well to confirm the quality of the results, since the FCS experiments support the major conclusions of this manuscript.

How sure are the authors that the obtained differences in diffusion times/coefficients are solely caused by differences in mobility of the protein (complexes) and are not affected by photobleaching. As stated in the manuscript (p13) the proteins bleach severely in WT cell but not in delta-yscD cells. Do the authors still observe (maybe less severe) bleaching in the delta-yscD cells? Have experiments been performed to test the effect of photobleaching at the acquired diffusion times, especially for slower moving proteins like eGFP-YscQ?

The authors state at page 13, line 347 that "Our data show that diffusion of the soluble components significantly differed between different data points for the same protein, revealing a wide range of complex compositions". However this last statement is not necessarily true since only a difference of mobility is observed with FCS without real confirmation that this would be caused by a change in composition.

Related to the point above: Are the authors pointing at the diffusion variation among the triplicate of measurements within one bacterium? If I understand correctly each bacterium was measured 3 times for 5 seconds. It is not clear to me if the diffusion times between the 3 sections varies a lot or is the variation mainly caused by the difference between the various bacteria.

I do not agree with the next statement (p13, line 348) "This indicates that the complexes are in constant exchange" because with FCS no exchange is observed unless this exchange is fast relative to the diffusion time. In this situation binding-unbinding could be included in the autocorrelation fitting model (Michelman-Ribeiro BiophysJ and others). I guess the authors want to state here that the composition is different between strains or are they pointing at the variation within the triplicate of measurements? If this later is the main point of focus it would be better to plot the variations among triplicates and not plot triplicates AND various bacteria measurements together.

Some minor issues:

- In Fig1B the localization of eGFP-tagged injectisome proteins has been visualized. When comparing the DIC with the fluorescence images for some of the YscQ cells no fluorescence is observed?
- The legend of Fig3B states that Box plots display single data points. (average transit time of 5 sec). Do I understand correctly that these points do not represent different individual bacteria, but the points represent a 5 second window analysis of a 3*5 sec measurement within one bacterium? So in total only one third of the number of points in the plot correspond to the number of

measured bacteria?

Reviewer #3 (Remarks to the Author):

In the manuscript entitled "A dynamic and adaptive network of cytosolic interactions governs export of proteins by the T3SS injectisome", Diepold et al. analyse the formation of cytosolic complexes by the soluble components YscN, YscL, YscQ and YscK of the T3SS from *Yersinia enterocolitica*. Using different fluorescence microscopy techniques and biochemical interaction studies, they show that all four proteins form complexes, which either associate with the T3SS or are present in the cytosol. Complex formation is affected by the extracellular calcium concentration, even in the absence of a functional T3SS. The authors suggest that this process is involved in the regulation of T3S, however, experimental evidence for this hypothesis is missing. The authors have previously described that the YscQ is present in a motile cytosolic and a T3SS-bound complex. A similar finding is described for YscN, YscL and YscK in the present study. The data presented for YscQ confirm the earlier results.

I have several comments and suggestions that are listed below:

Title and abstract:

It is not stated with which organism the experiments have been performed.

Line 135

The authors should explain in the text how they tested the functionality of the fusion proteins. If I understood correctly, the wild-type genes were replaced by fusion constructs encoding EGFP-tagged proteins in the genome.

General comment to the experiments:

It should be described how often the experiments were performed and if the results were reproducible.

Figure 1C

More explanations on microscopy techniques would be helpful. What are DIC and single colour micrographs? The abbreviation DIC is not explained in the manuscript. This is difficult to understand for readers without expertise in microscopy techniques.

Figure 1F

There is no scale on the X- and Y-axis.

Figure 1E

The abbreviation a.u. is not explained.

Line 178

The authors refer to Fig. S4 for the comparison of fluorescence intensities but no fluorescence is shown in Fig. S4.

Line 191

What do the authors mean with "these components"?

Lines 196-197

Here and throughout the text, more explanations on microscopy techniques could be provided to facilitate the understanding also for non-specialists. How can the fluorescence intensity be used to calculate the number of spots per bacterium? I thought the number of spots were counted?

Table 2

The authors should better explain how they quantify interactions by LC-MS/MS. Were negative controls included, i.e. were all interaction partners tested against the matrix alone to exclude unspecific binding? These data should be included as well.

Line 226: the terms "Halo-tagged strains" and "untagged controls" should be exchanged. I guess the authors refer to strains synthesizing Halo-tagged proteins.

Fig. 2C

This figure just shows two protein bands and is not very convincing. The authors should include negative controls. Do the bands correspond to YscQ or the smaller YscQ protein?

Fig. 2D

The scheme suggests that YscN does not interact directly with YscQ. If the interaction is indirect and requires YscL as is indicated, it would be interesting to perform interaction studies with YscN and YscQ in an yscL mutant.

Fig. 3C

Is the figure partially redundant with Fig. 3B? As in Fig. 3B, it should be indicated which proteins are EGFP-tagged.

Lines 342-343

Are these data shown? A reference is needed here.

Line 348

Again, this is difficult to understand for the non-specialist. How were the different data points measured? And what exactly is the transit time?

Line 472

What are the "different member components"?

Lines 522-525

This sentence is very difficult to understand. The authors speculate that the increased YscK diffusion leads to an increased exchange of YscQ and thus promotes effector export. However, in the next sentence, this hypothesis is stated as a fact. This should be corrected.

Fig. S1

I suppose that the genes encoding the EGFP fusions were inserted in the genome. This should be explained in the figure legend. Figures legends could be more informative throughout the manuscript. In Fig. S1, for example, it is not clear how the bacteria were cultivated and how the proteins were visualized. Is this a Coomassie staining? (Similar for Fig. S6).

Fig. S2

Were all fusion proteins stable and analysed by immunoblotting?

Fig. S4, right side

Was the blot also exposed for a longer time period? The contrast seems to be increased (grey background on top but not on the bottom of the film). In this case, it is difficult to visualize small proteins that could correspond to degradation products.

Reviewer #1 (Remarks to the Author):

In this study, the authors utilized some cutting edge technologies to demonstrate the interactions among the T3SS cytosolic proteins and differences in the assembly under inducing and non-inducing conditions, providing insights into the dynamic and adaptive interaction network of the soluble T3SS in the cytosol and its relation with the bound cytosolic complex at the injectisome. However, previous studies in *Yersinia*, *Salmonella* and *Shigella* on similar topics, addressed by conventional methods of protein-protein interaction, predicted most of the outcomes of the current study, thus the results are largely confirmatory in nature. The authors did make an important new observation, a type III needle independent sensing of the calcium for the intracellular complex formation, suggesting a previously unidentified mechanism for the regulation of type III secretion. Unfortunately, this was not pursued further in the current study.

We thank the reviewer for his/her comments and suggestions. We now characterize the influence of calcium on the cytosolic complex and the connection to effector secretion in greater detail with two additional datasets: (i) We combined expression and interaction data under secreting and non-secreting conditions. This allowed us to determine which pairwise interactions are influenced by to the shift in external conditions, and to which extent. We noticed that the interaction of YscQ with itself and with YscK were drastically decreased under secreting conditions, while other interactions, notably YscQ-YscN and YscL-YscL showed a much less pronounced or no decrease at all. This data, presented in the new **Fig. 2d** and **Table S3**, corroborates our FCS results, and provides an underlying molecular cause for the effect of extracellular calcium on the diffusion of the cytosolic components. (ii) We performed additional FCS experiments for the two central components of the cytosolic complex, YscL and YscQ, using different Ca^{2+} levels including intermediary concentrations around the critical Ca^{2+} concentration for effector secretion. In addition, we tested and quantified the level of secretion in a wild-type strain using the same conditions. Correlating the diffusion coefficients with the resulting secretion shows that diffusion and secretion display a linear correlation up to the point of maximal secretion. Beyond this point (upon addition of EDTA), secretion did not increase further, while the diffusion of the soluble components continued to increase (new **Fig. 3d** and **S11**). The new FCS data aligned well with the previous results; including the new data points hence also increased the statistical significance of the effect of extracellular Ca^{2+} in some cases (e.g. **Fig. 3c**).

Together, our new data suggest that diffusion of the soluble T3SS components is directly influenced by the extracellular conditions, most likely via changes in the affinity of cytosolic components. In turn, this (either the change in diffusion as such or the underlying change in the interaction network) allows effector secretion. Importantly, the diffusion of cytosolic GFP did not change upon changes in the extracellular Ca^{2+} level, indicating that the observed effects are specific for the studied T3SS components.

Minor points:

1. Fig. 2C, the protein levels in bacterial lysate and a negative control should be included. The size differences between YscQ and YscQc are not clear from the gel.

To address this request, we performed an additional small-scale purification. This allowed us to include an additional immunoblot using polyclonal YscQ antibody for total cellular protein (lysate) and the elution fraction of the two strains shown in **Fig. 2c**, as well as an additional control strain (ΔyscQ), in the

new **Fig. S7**. In addition, **Fig. 2c** itself now shows a much larger fraction of the blot, including the full-size Halo-YscQ band. This should make the size difference clearer.

2. Line 27, "..... regulation type III secretion.", missing "of".

Corrected.

3. Line 37, it is *exoU* gene, not the "functional T3SS", that is associated with increased antibiotic resistance.

Thanks for the remark. Indeed, reference 7 suggests that *exoU* is the predominant factor in increased antibiotic resistance, and a main indicator of bad clinical prognosis. We now mention this in the introduction.

Reviewer #2 (Remarks to the Author):

The authors studied the four soluble proteins involved in the injectosome. In this manuscript that is clearly written, the protein localisations are observed upon deletion of several proteins involved, performing co-immunoprecipitation experiments and FCS. Fluorescence microscopy showed that each of the soluble components localize to the injectosome, requiring the presence of all other partners and intensity analysis of the foci resulted in estimates of the relative stoichiometry of the injectosome complex. Coimmunoprecipitation revealed that all interactions were present, but significantly reduced in absence of YscD, indicating that the complexes in the cytosol are less stable than the injectosome-bound ones.

The microscopy and immunoprecipitation studies seem to be solid but I have some questions concerning the fluorescence fluctuation studies:

The authors tried to study the interaction of the soluble injectosome proteins using FCCS. Figure 3A shows the auto- and cross-correlation curves of a negative control and a YscQ and YscL sample. The negative sample clearly gives a near-zero amplitude as anticipated and the Ysc-sample a clear positive amplitude. However, as the authors mention in the main text the tagged mCherry protein is sensitive to photobleaching. A real danger in FCCS experiments is when one of the signals decreases strongly (e.g. due to photobleaching) and the other fluorescence signal decreases as well, maybe only slightly (minor bleaching). In this case a strong false-positive cross-correlation amplitude could be observed that could easily overwhelm a true low-amplitude cross-correlation signal. In Figure 1A-A it's now difficult to judge the quality of the cross-correlation signal since this blue graph has been plotted behind the autocorrelation curves. In order to see if the cross-correlation adds reliable info to the manuscript I would like to see the two original intensity traces and a different way of plotting the blue graph, such that the whole curve can be seen.

We fully agree with the reviewer's concern – he/she is addressing the main challenge of the F(C)CS experiments. We paid significant attention to avoid the false-positive FCCS due to photobleaching. Thus, we optimized our data acquisition to avoid data biased by photobleaching. We recorded data at very low excitation intensities (as detailed in one of our next answers). Therefore, we can confirm that the non-zero FCCS amplitude is definitely due to co-diffusion. As the Reviewer asked, we now added two-channel intensity traces to show that there is no bleaching for the curves we considered for analysis (new **Fig. S9b**). Also, we have re-drawn the blue (cross-correlation) curve as requested.

In addition, I question why the authors did not use FCCS to study all the interactions in the soluble injectosome complex because this could give a wealth of information about the composition of the complexes. The authors claim they are the first ones measuring FCCS in bacteria but I still doubt if these kind of measurements are possible in a reliable manner (see remark above) due to the limited amount of labeled protein, the small confined bacterial volume especially in combination with the problems of photobleaching of the photo-unstable yellow/orange/red fluorescent proteins. Is this also the reason the authors continue their analysis in the manuscript using only single colour FCS (using the more photostable eGFP)? As such the FCCS experiment shown here does not add much to the story.

As pointed out, it was quite challenging to get useful FCCS curves. Therefore the presented FCCS data aims as a proof-of-principle to show that the interaction we determined biochemically can also be observed with microscopy-techniques in live bacteria. We are certain that showing representative FCCS data is important to show that such experiments work in principle. Such experiments will become more and more probable with increasingly photostable and bright fluorescent proteins emerging. Nevertheless, we agree with the Reviewer that our FCCS data is quite limited; thus we now moved the FCCS part to the supplementary information (new **Fig. S9a**).

The authors use the diffusion time for a qualitative but also quantitative comparison between different strains. Although the qualitative comparison probably holds I have some doubts about the absolute diffusion coefficients presented in the article.

We agree with the reviewer, thus we now corrected the “diffusion coefficient” as “apparent diffusion coefficient” throughout the manuscript. It is worth noting that we are rather interested in relative changes in mobility, therefore absolute values of diffusion coefficients are of less importance to us than their relative alterations.

At page 22 the authors state they have used a 2-dimensional diffusion plus triplet model for fitting the autocorrelation curve. The equation of the full model or a reference should be included in the text.

We apologize for having missed out on this. We use the conventional fitting model which is now included it in the Methods section.

$$G(\tau) = \frac{1}{N} \left(1 + \frac{\tau}{\tau_D}\right)^{-1} \left[1 + T(1 - T)^{-1} \exp\left(\frac{-\tau}{\tau_{Tr}}\right)\right]$$

In addition, I wonder how accurate the obtained diffusion coefficients are since due to the limited height of the bacteria, intensity fluctuations and diffusion in the z-direction of the inhomogenous detection volume is most likely severely constrained but certainly occurring.

I guess in this case it would be better to use a constrained diffusion model with a limited z-profile (like presented by Generich et al (BiophysJ) and others) for fitting?

This would indeed be the perfect fitting model if we were interested in absolute values of diffusion coefficients. As pointed out, we are rather interested in relative changes in mobility, to help us elucidate which specific proteins and external conditions influence the state and function of the cytosolic T3SS components.

The diffusion coefficient we obtained for GFP is similar to literature values, thus we believe the absolute diffusion coefficients we obtained are fairly accurate. However, as pointed out before, our main interest is in the relative changes in mobility. Therefore we agree with the reviewer and we have added a comment clarifying this issue in the discussion.

It's probably a good idea to include a typical correlation curve including fit and fit-residuals in the supplementary info as well to confirm the quality of the results, since the FCS experiments support the major conclusions of this manuscript.

We agree that this is a good idea – we have added it as **Fig. S10a**. We would like to note that in case the fit quality was not good enough (i.e., when the $\chi^2 > 0.05$), we excluded the according FCS data. Our custom-written software tags the curves with bad quality, thus any bias is avoided. We now discuss this in the Methods section.

How sure are the authors that the obtained differences in diffusion times/coefficients are solely caused by differences in mobility of the protein (complexes) and are not affected by photobleaching. As stated in the manuscript (p13) the proteins bleach severely in WT cell but not in delta-yscD cells.

As the reviewer points out, photobleaching is in principle the main limitation of FCS measurements in bacteria. As mentioned before, we avoided extensive photobleaching by using low laser powers (as low as 1-2 μW), which is also the reason why the mCherry FCS data is relatively noisy. Since this point is quite important for potential scientists that would apply FCS on bacteria, we prepared a supplementary figure (**Fig. S10b**) showing representative FCS data suffering from photobleaching that should not be used for analysis.

Do the authors still observe (maybe less severe) bleaching in the delta-yscD cells? Have experiments been performed to test the effect of photobleaching at the acquired diffusion times, especially for slower moving proteins like eGFP-YscQ?

As mentioned, we use low laser powers to avoid extensive photobleaching and we only use accurate FCS data for analysis. As a further control, we recorded FCS data in bacteria and determined average transit times for different laser powers; a decrease in values of transit time for increasing laser powers is an indicator for photobleaching (as in detail pointed out in Clausen et al, Methods, 2015). As depicted in the graph below, we have not observed such a decrease within the laser power range used in this study. We have added this plot in our supplements (**Fig. S10c**).

The authors state at page 13, line 347 that “Our data show that diffusion of the soluble components significantly differed between different data points for the same protein, revealing a wide range of complex compositions”. However this last statement is not necessarily true since only a difference of mobility is observed with FCS without real confirmation that this would be caused by a change in composition.

Thanks for pointing this out. We have amended this statement: “compatible with a wide range of concurrent complex compositions”.

Related to the point above: Are the authors pointing at the diffusion variation among the triplicate of measurements within one bacterium? If I understand correctly each bacterium was measured 3 times for 5 seconds. It is not clear to me if the diffusion times between the 3 sections varies a lot or is the variation mainly caused by the difference between the various bacteria.

Sorry for having caused confusion. The main variation is between bacteria, which is now stated in the Results section.

I do not agree with the next statement (p13, line 348) "This indicates that the complexes are in constant exchange" because with FCS no exchange is observed unless this exchange is fast relative to the diffusion time. In this situation binding-unbinding could be included in the autocorrelation fitting model (Michelman-Ribeiro BiophysJ and others). I guess the authors want to state here that the composition is different between strains or are they pointing at the variation within the triplicate of measurements? If this later is the main point of focus it would be better to plot the variations among triplicates and not plot triplicates AND various bacteria measurements together.

This is a good point and we agree with the reviewer. However, in our opinion such detailed analysis is not required, since the main variation is between different bacteria not within one bacterium. We have clarified this further and amended the above sentence: "This suggests that the composition of complexes differs within a bacterial population, possibly caused by a constant turnover of interactions."

Some minor issues:

- In Fig1B the localization of eGFP-tagged injectisome proteins has been visualized. When comparing the DIC with the fluorescence images for some of the YscQ cells no fluorescence is observed?

This observation is correct. Irrespective of the labelled protein, we observe a small proportion (usually <5% of bacteria) with no visible fluorescence. In many of these cases, the cell shape does not indicate any possible reason, and we did not notice any correlation of the presence of these bacteria with external stimuli or experimental conditions. We now mention this fact in the Results section.

- The legend of Fig3B states that Box plots display single data points. (average transit time of 5 sec). Do I understand correctly that these points do not represent different individual bacteria, but the points represent a 5 second window analysis of a 3*5 sec measurement within one bacterium? So in total only one third of the number of points in the plot correspond to the number of measured bacteria?

That is correct, although the number of bacteria is higher than that, because we discarded some measurements which did not meet the quality criteria. The reviewer's point is now explicitly indicated in the figure legend.

Reviewer #3 (Remarks to the Author):

In the manuscript entitled "A dynamic and adaptive network of cytosolic interactions governs export of proteins by the T3SS injectisome", Diepold et al. analyse the formation of cytosolic complexes by the soluble components YscN, YscL, YscQ and YscK of the T3SS from *Yersinia enterocolitica*. Using different fluorescence microscopy techniques and biochemical interaction studies, they show that all four proteins form complexes, which either associate with the T3SS or are present in the cytosol. Complex formation is affected by the extracellular calcium concentration, even in the absence of a functional T3SS. The authors suggest that this process is involved in the regulation of T3S, however, experimental evidence for this hypothesis is missing.

The authors have previously described that the YscQ is present in a motile cytosolic and a T3SS-bound complex. A similar finding is described for YscN, YscL and YscK in the present study. The data presented for YscQ confirm the earlier results.

As mentioned in the reply to reviewer 1, we now characterize the influence of calcium on the cytosolic complex and the connection to effector secretion in greater detail with two additional datasets.

For the first part, the link between extracellular calcium levels and the diffusion rates of the soluble components, we corrected pairwise protein interactions measured under secreting and non-secreting conditions for the differences in protein expression under these conditions (new **Fig. 2d** and **Table S3**). This shows which interactions are influenced most by to the shift in external conditions. In line with our FCS results, the interactions of YscQ with itself and with YscK were drastically decreased under secreting conditions (**Fig. 2d**). This highlights a possible underlying molecular cause for the effect of extracellular calcium on the diffusion of the cytosolic components.

To study the second part, the connection between the cytosolic complexes and secretion, in more detail, we performed additional FCS experiments, including Ca²⁺ levels close to the "tipping point" for secretion. We then correlated the measured diffusion levels for YscQ and YscL and linked them to the secretion levels of a wild-type strain under the same conditions (new **Fig. 3d** and **S11**). We found that diffusion and secretion display a linear correlation up to the point of maximal secretion. Further addition of EDTA still showed an influence on diffusion, while it did not further increase secretion. Diffusion of GFP, used as a control, was not influenced by the external calcium levels.

Together, our new data provide a link between our observations, and the physiological outcome (effector secretion) and suggests that diffusion of the soluble T3SS components is directly influenced by the extracellular conditions, most likely via changes in the affinity of cytosolic components. In turn, this (either the change in diffusion as such or the underlying change in the interaction network) allows effector secretion.

I have several comments and suggestions that are listed below:

Title and abstract:

It is not stated with which organism the experiments have been performed.

Thank you. We now mention in the abstract (line 21) that the experiments were performed in *Yersinia enterocolitica*.

Line 135

The authors should explain in the text how they tested the functionality of the fusion proteins. If I understood correctly, the wild-type genes were replaced by fusion constructs encoding EGFP-tagged proteins in the genome.

That's correct. We now briefly explain the construction of the strains and the functionality assay in this part of the Results section, and refer to the more detailed description in the Methods section.

General comment to the experiments:

It should be described how often the experiments were performed and if the results were reproducible.

We now include this information in the respective figure legends and/or the Methods section, and the exact number of data points in the new **Fig. S4**, where required.

Figure 1C

More explanations on microscopy techniques would be helpful. What are DIC and single colour micrographs? The abbreviation DIC is not explained in the manuscript. This is difficult to understand for readers without expertise in microscopy techniques.

Thanks for the remark. DIC stands for differential interference contrast, a widely used optical microscopy technique used to enhance contrast in transparent samples without staining. Single colour referred to either GFP or mCherry (as opposed to the overlays) shown in the lower part of **Fig. 1C**. The figure legend now indicates this. In general, we have included a short basic description of the microscopy techniques in the Results section (in addition to the introduction), and tried to explain the background of the microscopy results in more detail throughout the Results section. In some cases, we also expanded the method descriptions of the fluorescence microscopy experiments.

Figure 1F

There is no scale on the X- and Y-axis.

The y-axis is normalized for the maximal fluorescence intensity in each trace; the x-axis is normalized for the bacterial diameter in each trace, with the marks showing the position of the fluorescence maxima at the bacterial membrane. This was previously only indicated in the figure legend; we now relabelled the axis labels and included a y-axis indicating the maximum.

Figure 1E

The abbreviation a.u. is not explained.

This is now explained as arbitrary units (raw fluorescence intensity data of the micrographs).

Line 178

The authors refer to Fig. S4 for the comparison of fluorescence intensities but no fluorescence is shown in Fig. S4.

Apologies, this refers to **Fig. S5**; **Fig. S4** shows the stability and functionality of the fusions to YscD and YscV. This has now been corrected.

Line 191

What do the authors mean with "these components"?

The soluble T3SS components. We've indicated this more clearly now.

Lines 196-197

Here and throughout the text, more explanations on microscopy techniques could be provided to facilitate the understanding also for non-specialists. How can the fluorescence intensity be used to calculate the number of spots per bacterium? I thought the number of spots were counted?

Thanks for pointing out this mistake! The spots were detected automatically, based on the fluorescence intensity above the background, and the shape of the fluorescent spot (required to be roughly round and in the size-range of a diffusion-limited spot), using the Imaris software (all details are listed in the Methods section). The fluorescence intensity of spots in strains expressing fusions to different T3SS components was then used to estimate the stoichiometry. This is now correctly stated in the table legend.

As mentioned above, we tried to explain the microscopy techniques in more detail throughout.

Table 2

The authors should better explain how they quantify interactions by LC-MS/MS. Were negative controls included, i.e. were all interaction partners tested against the matrix alone to exclude unspecific binding? These data should be included as well.

As a negative control for the interaction studies and the quantification by LC-MS/MS, we have used an untagged *Y. enterocolitica* strain grown under the same conditions. YscK was not detected in the negative control at all, while YscL, N and Q were detected, were strongly enriched in the tagged strains. These results are contained in **Table 2**.

We now include this information as well as a more detailed protocol of the LC-MS/MS quantification in the Methods part (subheading "HaloTag-based interaction studies, mass spectrometry and quantification"), as requested.

Line 226: the terms "Halo-tagged strains" and "untagged controls" should be exchanged. I guess the authors refer to strains synthesizing Halo-tagged proteins.

We thank the reviewer for the remark, and have replaced "Halo-tagged strains" and "untagged controls" by "strains expressing the respective Halo-tagged proteins" and "WT control strains".

Fig. 2C

This figure just shows two protein bands and is not very convincing. The authors should include negative controls. Do the bands correspond to YscQ or the smaller YscQ protein?

As mentioned in our reply to reviewer 1, we now include the complete blot in **Fig. 2c**, which also shows the band for full-size Halo-YscQ and the sizes of the different versions of YscQ. In addition, we have performed an additional co-IP experiment which allowed us to now include an additional immunoblot (**Fig. S7**) using polyclonal YscQ antibody for total cellular protein (lysate) of the two strains shown in **Fig. 2c** and a control ($\Delta yscQ$).

Fig. 2D

The scheme suggests that YscN does not interact directly with YscQ. If the interaction is indirect and requires YscL as is indicated, it would be interesting to perform interaction studies with YscN and YscQ in an *yscL* mutant.

That would be interesting indeed! We tried to address the question by constructing a YscL mutant in an mCherry-YscN/EGFP-YscQ FCCS strain. Unfortunately, the high bleaching rate of mCherry-YscN prevented a reliable analysis. For this reason, we rely on previous studies for our reasoning (and the scheme).

While it clear that YscQ and YscN are part of a protein complex, there is conflicting evidence on a **direct** interaction of YscN and YscQ and their respective homologues in other T3SS. Whereas Biemans-Oldehinkel et al. (J Bact, 2011) detected interaction of the (overexpressed) EPEC homologues of YscN and YscQ in co-IP experiments, the Yeast-two-hybrid and Yeast-three-hybrid analysis of Jackson and Plano (FEMS Microbiol Lett, 2000) showed a requirement of YscL for the interaction of YscN and YscQ, and no direct interaction between the two proteins was detected by Jouihri et al. (Mol Micro, 2003) or in other Y2H screens. The flagellar data is similarly unclear. We now directly indicate this uncertainty in the figure legend.

Fig. 3C

Is the figure partially redundant with Fig. 3B? As in Fig. 3B, it should be indicated which proteins are EGFP-tagged.

This figure is redundant with Fig. 3B (now **Fig. 3a**) only as far as the controls for the deletion strains (the narrow boxplot-only lanes showing the respective WT and GFP transit times) were included to allow an easy identification of the impact of the deletions.

We originally had figure 3B and 3C (now **Fig. 3a** and **3b**) combined, but test readers found it very difficult to extract the information. The current setup also allows to distinguish between the differences between diffusion of different proteins (**3a**) and the impact of the deletions in the respective strains (**3b**).

We thank the reviewer for pointing out the inconsistent nomenclature of the EGFP fusion proteins in panels B and C, we have now fixed that.

Lines 342-343

Are these data shown? A reference is needed here.

We now show the comparison of WT and $\Delta yscD$ strains of EGFP-YscQ and EGFP-YscL as a new supplementary figure (**Fig. S10de**).

Line 348

Again, this is difficult to understand for the non-specialist. How were the different data points measured? And what exactly is the transit time?

The transit time is the time a fluorescent molecule is detected, i.e. the time it resides within the microscopy focal area. Fast-diffusing (i.e. small) molecules diffuse in and out of this area more quickly than slow-diffusing (i.e. large or tethered) molecules. The transit time is hence directly negatively correlated with the diffusion coefficient.

Experimentally, the transit time is detected by measuring the fluorescence intensity in the focal area at a high rate, and calculating the autocorrelation function (the similarity between the fluorescence intensity at t and $t+\Delta t$, displayed now in the new **Fig. S10a**). The inflection point of the fit for this autocorrelation curve (or in other words, the Δt at which on the fluorescence intensity is half-maximally similar to the original intensity) is the average transit time for the respective data series (marked by an arrow in **Fig. S10a**). In our experiments, each data point represents the average transit time of a five-second measurement.

Apologies for being unclear at this point. We have now included a short paragraph describing these basic principles of FCS in the Results section. The experimental details are included in the Methods section.

Line 472

What are the "different member components"?

The members of the cytosolic complexes themselves, the soluble T3SS components Sct/YscK,Q,L,N. This is now mentioned explicitly.

Lines 522-525

This sentence is very difficult to understand. The authors speculate that the increased YscK diffusion leads to an increased exchange of YscQ and thus promotes effector export. However, in the next sentence, this hypothesis is stated as a fact. This should be corrected.

We thank the reviewer for this remark. We have tried to structure this idea more clearly and indicate its speculative nature throughout. The paragraph now reads "This might lead to the increased exchange of YscQ between the injectisome and the cytosolic pool observed earlier, and in turn allow effector export by the T3SS. Such a mechanism would be unique to the T3SS and not be required in the flagellum. Intriguingly, SctK/YscK, which we propose to act as sensor or transducer of this signal, is the only soluble T3SS component that does not have a flagellar homologue."

Fig. S1

I suppose that the genes encoding the EGFP fusions were inserted in the genome. This should be explained in the figure legend. Figures legends could be more informative throughout the manuscript. In

Fig. S1, for example, it is not clear how the bacteria were cultivated and how the proteins were visualized. Is this a Coomassie staining? (Similar for Fig. S6).

As mentioned earlier, the wild-type genes on the virulence plasmid were replaced by N-terminal fusions of EGFP to the respective gene by allelic exchange. This is now explicitly stated at the start of the Results section. In **Figures S1, S4 and S6**, we now indicate that the gels display Coomassie-stained of TCA-precipitated proteins in the culture supernatant after three hours of incubation under secreting conditions. Details about the culture conditions are given in the Methods section (subheading “*Y. enterocolitica* cultures for secretion and microscopy analysis”).

Fig. S2

Were all fusion proteins stable and analysed by immunoblotting?

The stability of all fusion proteins was analysed by immunoblotting. All proteins were >90% stable with minor degradation bands about 15-20 kDa below the expected molecular weight, as published earlier (Diepold et al., PLOS Biol 2015), with the exception of YscN, which shows a more prominent degradation band in addition to the full-size band. Importantly, no band corresponding to free EGFP (25-30 kDa) could be detected for any strain.

The data is now included as new **Fig. S1B** and referred to in the Results section.

Fig. S4, right side

Was the blot also exposed for a longer time period? The contrast seems to be increased (grey background on top but not on the bottom of the film). In this case, it is difficult to visualize small proteins that could correspond to degradation products.

We did not apply any non-linear contrast enhancement in this figure (or, indeed, any other figure in the manuscript). For the information of the reviewer and the editor, we show the raw data of the longest exposure of this immunoblot (14 min) below and as a separate attachment (lanes from left to right: WT, lower amount of EGFP-YscD, EGFP-YscD, YscV-EGFP, the last two lanes are displayed in **Fig. S4**).

Reviewers' comments:

Reviewer #1 (Remarks to the Author):

The authors utilized a cutting edge technology that was well described in eukaryotic systems to address a significant question in bacteria, i.e. type III secretion system. Unfortunately, most of the reported findings of this manuscript are not new, they are confirmative in nature. Although the authors addressed some of the concerns this reviewer had, especially on the effect of calcium on the intracellular complex formation, their new findings are not significant enough to warrant publication in this prestigious journal.

The authors described many observations in terms of protein-protein interactions, however, most of those lack direct biological relevance, i.e. the function of type III secretion system. For instance, the authors described that "the soluble components form a large complex at the proximal interface of the injectisome" which would confirm previous studies, but the significance of this observation is not clear. Also, the authors describe "slow or fast diffusion of the soluble components", without quantitative analysis, again biological significance of their observations has not been elaborated.

Reviewer #2 (Remarks to the Author):

The authors correctly address all the suggestions and issues I have raised in my previous review in their rebuttal letter. Especially the discussion about the potential photobleaching artifacts in the F(C)CS experiments and the addition of supplemental Fig S10 strengthens the paper. Also the other additions and changes in the main text have improved the manuscript significantly and therefore I would like to recommend to accept this paper for publication in Nature Communications.

Reviewer #3 (Remarks to the Author):

The revised version of the manuscript entitled "A dynamic and adaptive network of cytosolic interactions governs protein export by the T3SS injectisome" by Diepold et al. is now significantly improved and the authors addressed all reviewer comments and changed the manuscript accordingly. The influence of extracellular calcium concentrations on protein-protein interactions and diffusion rates of single proteins is now analysed in more detail. However, I have a few questions concerning the effect of extracellular calcium levels which are listed below:

The authors state in the discussion (line 455/6) that the effect of external calcium levels is even observed in strains lacking YscD. I did not find the data supporting this statement. In which figure was this shown? In lines 482 – 488, the authors then speculate that the chelation of calcium weakens the interaction between YscK and YscQ, which might lead to an increased exchange of YscQ. However, if the effect of calcium chelation is also observed in an yscD mutant, the interaction between YscK and YscQ is already weakened in this strain as is shown in Fig. 2B.

In Fig. 2D, the authors now analysed protein-protein interactions under secreting and non-secreting conditions. The interaction between Halo-YscQ and YscL is increased under non-secreting conditions according to this figure. However, in Table 2, the interaction between Halo-YscL and YscQ is reduced under non-secreting conditions if I understand this table correctly. Isn't this contradictory? Why did the authors not perform the experiments in Fig. 2D with the same Halo-tagged proteins but use Halo-YscQ instead of Halo-YscL?

I also do not really understand Fig. 3D. How can the secretion in the wild-type strain be correlated to transit times of EGFP fusion proteins in another strain? At least, this is the description of the experimental setup in the answer of the authors to the reviewer comments. In the figure legend for Fig. 3D, it is not explained which strains exactly were used. Furthermore, there is no scale on the X axis. If the transit times and secretion efficiencies were analysed at specific calcium concentrations, the data points cannot be connected with lines.

I also have a couple of minor points that are listed below:

Table 1:

With which strains were these experiments performed?

Legend of Table 2:

The term " pull-downs by Halo Tag-fused YscL and YscQ" should be rephrased. The authors still used the term "tagged strains" (lines 746/7, see also line 308)

Figure 1F

I am not sure if I understand this figure correctly. The two peaks reflect the highest fluorescence intensities for each of the proteins, suggesting that they are present in two spots in the bacterium. But there are cells with more than two foci. The X axis is not very informative. Does this mean that all foci have approximately the same distance to each other?

Lines 137/138

How can the genes be replaced by "N-terminal fusions of EGFP to the respective genes"?

Line 140

The sentence starting in line 140 is very long and difficult to follow (similar for the sentences starting in line 152 and line 321).

Line 188

It is not clear from the text with which proteins YscK, Q, L and N interact.

Line 596

The text " calculated spectral index for that protein in the pull-down for that strain to that of the untagged control" should be corrected

Reviewer #1 (Remarks to the Author):

The authors utilized a cutting edge technology that was well described in eukaryotic systems to address a significant question in bacteria, i.e. type III secretion system. Unfortunately, most of the reported findings of this manuscript are not new, they are confirmative in nature. Although the authors addressed some of the concerns this reviewer had, especially on the effect of calcium on the intracellular complex formation, their new findings are not significant enough to warrant publication in this prestigious journal.

The authors described many observations in terms of protein-protein interactions, however, most of those lack direct biological relevance, i.e. the function of type III secretion system. For instance, the authors described that “the soluble components form a large complex at the proximal interface of the injectisome” which would confirm previous studies, but the significance of this observation is not clear. Also, the authors describe “slow or fast diffusion of the soluble components”, without quantitative analysis, again biological significance of their observations has not been elaborated.

While the main focus of this study was on the interactions between components under different conditions, which directly links to their function, the diffusion rates of the cytosolic components were analysed quantitatively. Indeed the determined diffusion times directly translate into apparent diffusion coefficients for the cytosolic components, and this is the first description of its kind. Previous estimates have in all *in vitro* and this is the first *in vivo* measurement. We now highlight this point more clearly in the Results section (lines 278-280 in the revised manuscript). The new regulatory concept uncovered by our study – adaptation of the cytosolic complexes to external conditions and its link to secretion – is already clearly pointed out in the Discussion (lines 333-336, 470-472, and 478-493).

Reviewer #2 (Remarks to the Author):

The authors correctly address all the suggestions and issues I have raised in my previous review in their rebuttal letter. Especially the discussion about the potential photobleaching artifacts in the F(C)CS experiments and the addition of supplemental Fig S10 strengthens the paper. Also the other additions and changes in the main text have improved the manuscript significantly and therefore I would like to recommend to accept this paper for publication in Nature Communications.

We thank the reviewer for his/her comments!

Reviewer #3 (Remarks to the Author):

The revised version of the manuscript entitled "A dynamic and adaptive network of cytosolic interactions governs protein export by the T3SS injectisome" by Diepold et al. is now significantly improved and the authors addressed all reviewer comments and changed the manuscript accordingly. The influence of extracellular calcium concentrations on protein-protein interactions

and diffusion rates of single proteins is now analysed in more detail. However, I have a few questions concerning the effect of extracellular calcium levels which are listed below:

The authors state in the discussion (line 455/6) that the effect of external calcium levels is even observed in strains lacking YscD. I did not find the data supporting this statement. In which figure was this shown?

The data presented in Fig. 3 is measured in a $\Delta yscD$ background. While this does not change the diffusion rate of the cytosolic complexes (as shown in Fig. S10de), it allowed to focus on the cytosolic complexes (see scheme in Fig. 2a) and the comparison between secreting and non-secreting conditions without the additional complications caused by the feedback regulation upon secretion. This is stated in the Results (lines 262-270) and the legend for Fig. 3a-c, but to make the point more clearly, we now also state this fact in line 271 and the legend for Fig. 3d.

In lines 482 – 488, the authors then speculate that the chelation of calcium weakens the interaction between YscK and YscQ, which might lead to an increased exchange of YscQ. However, if the effect of calcium chelation is also observed in an $yscD$ mutant, the interaction between YscK and YscQ is already weakened in this strain as is shown in Fig. 2B.

This observation is correct and highlights an important point. The difference between WT and $\Delta yscD$ strains is that there are no injectisome-bound cytosolic complexes in the absence of YscD (see scheme in Fig. 2a). The decreased interaction between YscK and YscQ in the absence of YscD therefore shows that injectisome-bound cytosolic complexes are more stable than free-diffusing complexes, as pointed out in lines 202-205 and discussed in lines 380-387.

The data in Fig. 3 focuses on the cytosolic complexes (by analysing $\Delta yscD$ strains) and shows an **additional** effect of the extracellular Ca^{2+} level on the diffusion of these free-diffusing complexes. We now point out this distinct effect more clearly in lines 481/482 and 488.

In Fig. 2D, the authors now analysed protein-protein interactions under secreting and non-secreting conditions. The interaction between Halo-YscQ and YscL is increased under non-secreting conditions according to this figure. However, in Table 2, the interaction between Halo-YscL and YscQ is reduced under non-secreting conditions if I understand this table correctly. Isn't this contradictory? Why did the authors not perform the experiments in Fig. 2D with the same Halo-tagged proteins but use Halo-YscQ instead of Halo-YscL?

Table 2 shows that interactions between the cytosolic components can be specifically measured by our approach. The data in Table 2 compare the mass spec detection levels of purified interacting proteins with the background levels detected in unlabelled strains under the same conditions, and do not allow to directly compare degrees of interaction between different strain backgrounds. For the remainder of our study, we focused on the Halo-YscQ strains, because the important deletion mutants were present for these strains (see Fig. 2b). Subsequently, we performed the large quantitative interaction experiment with these strains.

I also do not really understand Fig. 3D. How can the secretion in the wild-type strain be correlated to transit times of EGFP fusion proteins in another strain? At least, this is the description of the

experimental setup in the answer of the authors to the reviewer comments. In the figure legend for Fig. 3D, it is not explained which strains exactly were used. Furthermore, there is no scale on the X axis. If the transit times and secretion efficiencies were analysed at specific calcium concentrations, the data points cannot be connected with lines.

Thanks for pointing out that our figure legend was imprecise at this point. To address the connection between the diffusion of the soluble T3SS components and effector secretion (as suggested by reviewers 1 and 3), we tested the influence of external Ca^{2+} levels (i) on effector secretion (measured in a wild-type strain, see top part of Fig. 3d and black line in bottom part), and (ii) the diffusion of two soluble components (EGFP-YscQ and EGFP-YscL, both in ΔyscD background, see red and orange lines in bottom part). Both experiments were performed at 5 mM CaCl_2 , 1 mM CaCl_2 , addition of neither CaCl_2 nor EDTA, and 5 mM EDTA. The labels in the centre therefore are valid for both the upper and lower part of the figure. We have now repeated the scale for the lower x axis, and removed the connecting lines in the lower part of the figure, as requested by the reviewer. In addition, we rewrote the figure legend and respective part of the results section (line 308) to make this point easier to understand.

I also have a couple of minor points that are listed below:

Table 1:

With which strains were these experiments performed?

These experiments were performed with the EGFP-labelled T3SS components used in Fig. 1be. We now indicate this in the table itself and the legend.

Legend of Table 2:

The term "pull-downs by Halo Tag-fused YscL and YscQ" should be rephrased. The authors still used the term "tagged strains" (lines 746/7, see also line 308).

Apologies for missing these jargon expressions. We now replaced them by "pull-down experiments using strains expressing Halo-YscL or Halo-YscQ, respectively" and "interaction ratio in strain expressing the Halo-tagged protein / WT control". Similarly, we corrected "Halo-tagged strains" in line 308 (now line 313/314).

Figure 1F

I am not sure if I understand this figure correctly. The two peaks reflect the highest fluorescence intensities for each of the proteins, suggesting that they are present in two spots in the bacterium. But there are cells with more than two foci. The X axis is not very informative. Does this mean that all foci have approximately the same distance to each other?

For this figure, we generated line profiles of fluorescence intensity across single bacteria. Each line used for the profile crossed two fluorescent foci on opposite sides of a bacterium. A scheme of this approach is now included in Fig. 1f. As the foci were not equidistant, we normalised the spot-to-spot distance to make the profiles comparable. The x-axis therefore spans from the fluorescence intensity maximum on one side of the bacterium (now labelled "0") to the intensity maximum on the other side (now labelled "1"). We hope that the updated figure and legend make this point clearer.

Lines 137/138

How can the genes be replaced by "N-terminal fusions of EGFP to the respective genes"?

We changed this sentence to "The wild-type genes on the virulence plasmid were modified by allelic exchange to encode for N-terminally EGFP-tagged versions of the respective protein".

Line 140

The sentence starting in line 140 is very long and difficult to follow (similar for the sentences starting in line 152 and line 321).

Thanks for this remark. In all cases, we have shortened the sentences, and additionally rephrased the sentences starting in lines 152 (153 in the revised version of the manuscript) and 321 (now 326), to make them easier to understand.

Line 188

It is not clear from the text with which proteins YscK, Q, L and N interact.

The proteins interact with each other. We have now added this information.

Line 596

The text "calculated spectral index for that protein in the pull-down for that strain to that of the untagged control" should be corrected.

We have replaced this expression by as follows: "Table 2 indicates the enrichment factor of detected interacting proteins in the strains expressing Halo-YscL or Halo-YscQ, over the untagged WT control strains, as determined by the ratio of their spectral indices, a measure of protein representation in each sample."

REVIEWERS' COMMENTS:

Reviewer #3 (Remarks to the Author):

The authors have addressed all my suggestions and questions. I have no additional comments to this manuscript.